 TOOLS AND RESOURCES

# On demand expression control of endogenous genes with DExCon, DExogron and LUXon reveals differential dynamics of Rab11 family members

Jakub Gemperle*, Thomas S Harrison, Chloe Flett, Antony D Adamson, Patrick T Caswell*

Wellcome Trust Centre for Cell-Matrix Research, School of Biological Sciences, Faculty of Biology Medicine and Health, Manchester Academic Health Science Centre, The University of Manchester, Manchester, United Kingdom

**Abstract** CRISPR technology has made generation of gene knock-outs widely achievable in cells. However, once inactivated, their re-activation remains difficult, especially in diploid cells. Here, we present DExCon (Doxycycline-mediated endogenous gene Expression Control), DExogron (DExCon combined with auxin-mediated targeted protein degradation), and LUXon (light responsive DExCon) approaches which combine one-step CRISPR-Cas9-mediated targeted knockin of fluorescent proteins with an advanced Tet-inducible TRE3GS promoter. These approaches combine blockade of active gene expression with the ability to re-activate expression on demand, including activation of silenced genes. Systematic control can be exerted using doxycycline or spatiotemporally by light, and we demonstrate functional knock-out/rescue in the closely related Rab11 family of vesicle trafficking regulators. Fluorescent protein knock-in results in bright signals compatible with low-light live microscopy from monoallelic modification, the potential to simultaneously image different alleles of the same gene, and bypasses the need to work with clones. Protein levels are easily tunable to correspond with endogenous expression through cell sorting (DExCon), timing of light illumination (LUXon), or by exposing cells to different levels of auxin (DExogron). Furthermore, our approach allowed us to quantify previously unforeseen differences in vesicle dynamics, transferrin receptor recycling, expression kinetics, and protein stability among highly similar endogenous Rab11 family members and their colocalization in triple knock-in ovarian cancer cell lines.

*For correspondence:
jakub.gemperle@gmail.com (JG);
patrick.caswell@manchester.ac.uk (PTC)

Competing interest: The authors declare that no competing interests exist.

## Editor's evaluation

Here the authors present a genome editing strategy that enables blocking and tetracycline-controlled, re-expression of fluorescently-tagged genes from endogenous loci. The authors combine this with a photoactivatable, tet-on/off system, a knocksideways approach, and the auxin-inducible degron system to improve spatial and temporal control of gene expression. These powerful tools are used to evaluate the localization, function and protein-expression dynamics of the Rab11 family of small GTPases.

## Introduction

The ability to control gene expression and analyze protein coding gene function inside cells is a critical step in our understanding of normal physiology and disease pathology. CRISPR/Cas9 technology has revolutionized gene editing approaches and makes generation of gene knock-outs or

knock-ins straightforward (*Doudna and Charpentier, 2014*). Combination of CRISPR with single-stranded DNA (ssDNA) donors has proven to be the most precise and efficient method of on-target integration (*Quadros et al., 2017*; *Li et al., 2017*), although knock-in approaches still suffer from low efficiencies, especially when aiming to achieve homozygosity (*Gurumurthy et al., 2019*). Fluorescent protein (FP) knock-in at endogenous loci leading to protein-fusion allows functional enrichment of complete knock-in events and study of native protein function using fluorescent microscopy (*Roberts et al., 2017*; *Leonetti et al., 2016*). However, they only complement loss-of-function/gain-of-function methods, and their usefulness is limited by brightness of the chosen FP. Conditional loss-of-function methods mainly rely on CRISPR/Cas9 combination with FLP-frt or Cre-LoxP systems (*Quadros et al., 2017*). However, these are labor intensive and mediated by recombination, which is by nature irreversible thus complicating attempts to rescue gene expression. In addition, to understand the full spectrum of gene functions, it is advantageous to reversibly tune expression levels, visualized by fluorescence microscopy, in a spatiotemporal context and not only depend on binary on/off approaches. Such levels of control can be partly offered by the auxin-inducible degron (AID) system (*Nishimura et al., 2009*) introduced as a tag into the endogenous locus via CRISPR/Cas9 (*Röth et al., 2019*). Here, an exogenously expressed plant auxin receptor F-box protein triggers ubiquitylation and degradation of target proteins fused to an AID tag on addition of the plant hormone auxin. However, this technology has its own limitations and caveats; for example, poor spatial control due to diffusion of auxin, significant basal degradation without auxin, and/or incomplete auxin-induced protein depletion.

Ideally, gene expression control and visualization of its protein product should be generated by a single gene modification (e.g. CRISPR knockin) at the endogenous locus and allow temporal, conditional, and reversible gene inactivation of all protein-coding transcripts, on demand, without the drawbacks of AID system. Such an approach is particularly useful for studying highly similar protein coding genes where their complementary and specific functions remain poorly understood, such as for members of Rab11 family of small GTPases: Rab11a, Rab11b, and Rab25 (Rab11c). Rab11s share high amino acid identity (Rab11a:Rab11b 89%, Rab11a:Rab25 66%, Rab11b:Rab25 61%), are known to play key roles in membrane transport, localize to recycling endosomes, and have been identified as important players in the cellular basis of an ever-increasing number of human disorders, including cell migration/invasion and cancer (*Wilson et al., 2018*; *Kelly et al., 2012*). Several Rab11 family isoforms have been reported (*Ota et al., 2004*), and specific antibodies are hard to obtain without cross-reactivity. Moreover, much what is known about their regulation and dynamics in live cells has been achieved with the aid of transiently transfected cDNA (complementary DNA)-mediated overexpression which may lead to non-physiological artifacts (*Ratz et al., 2015*; *Gibson et al., 2013*).

To overcome the limitations of previously established gene expression control technologies, we generated tractable single CRISPR knock-in based strategies for tunable and reversible gene expression control capable of knock-out/rescue and endogenous gene re-activation. These modifications are ideal for microscopy when combined with FP knock-in, are free of transfection artifacts, preserve post-transcriptional and post-translational control, and avoid comprehensive and labor-intensive genotyping post-editing of clones. Specifically, we harnessed CRISPR/Cas9 technology for targeted knock-in of the third-generation tetracycline-inducible system (Tet-On 3G) promoter TRE3GS followed by an FP coding region. This promoter lacks binding sites for endogenous mammalian transcription factors, is effectively silent in the absence of induction, and any transcription driven by endogenous active promoter generates non-sense frameshift. Simultaneous or sequential silencing of additional alleles was achieved by additional ssDNA donor coding antibiotic resistance to attain complete and selectable functional knock-out. Spatiotemporal control was achieved by combination with a photoactivatable (PA)-Tet-OFF/ON system (*Yamada et al., 2018*), even in cells within 3D microenvironments. We further established a tunable dual-input system combining TetON and AID, termed DExogron (DExCon [Doxycycline-mediated endogenous gene Expression Control] combined with auxin-mediated targeted protein degradation), that enhanced switch off kinetics of the TetON system, allowing rapid complete protein depletion or adjustment of expression to physiological levels on demand.

Our optimized pipeline to deliver this suite of CRISPR tools was applied to modulate endogenous gene expression of Rab11 family members, demonstrating that re-activation of endogenous Rab25 expression can promote the invasive migration of cancer cells, and revealing new insight into the localization and functions of Rab11a, Rab11b, and Rab25.

## Results

Deconvolving the relative functions of genes within highly related gene families has proven difficult, for example, the Rab11 family has a complex relationship with cancer, with several potential oncogenic and tumor suppressive functions identified (*Kelly et al., 2012*; *Jin et al., 2021*; *Cheng et al., 2004*). However, gene editing offers potential solutions. In order to identify the best knockin strategies to visualize the Rab11 gene family, we first analyzed the expression patterns of the Rab11s across tissues and in cancer using UCSC Xena, an online exploration tool for multi-omic data (*Goldman et al., 2020*). Rab11a, Rab11b, and Rab25 were significantly enriched in ovarian cancer, both for primary and recurrent tumors (*Figure 1—figure supplement 1A*). Bioinformatic analyses suggested the existence of additional potentially protein-coding human Rab11a/b splice variants (*Figure 1—figure supplement 1B*). Analysis of expression quantitative trait loci (eQTLs) revealed abundant expression of two Rab11b-coding and five Rab11a-coding transcripts across multiple healthy tissues while Rab25 is more restricted, but expression is increased across multiple types of cancer (*Figure 1—figure supplement 1C*, https://doi.org/10.48420/16988617). Similarly, several Rab11a/Rab11b/Rab25 splicing isoforms are enriched in ovarian cancer compared to normal tissue (*Figure 1—figure supplement 1D*). All significantly expressed transcripts share an intact N-terminus; however, for some splice variants, amino acids important for GTP/GDP binding (which allow oscillation between active/inactive states) or for the attachment to the outer leaflet of endosomal membranes are missing, which could therefore profoundly alter Rab11s function (*Figure 1—figure supplement 1E*). Thus, to capture the full physiological picture of the combination of Rab11s protein-coding transcripts, we fluorescently tagged the N-terminus of Rab11s using CRISPR knockin (*Figure 1A*). We used long ssDNA donor for precise and efficient on-target integration (*Quadros et al., 2017*; *Li et al., 2017*; *Miura et al., 2018*) with flanking homologous arms 150–300 bases prepared via RNA intermediates (*Figure 1B*). Initial testing of different approaches led to our optimized protocol (*Figure 1C*) combining magnetofection of long ssDNA and high fidelity (Hifi) Cas9 delivered as ribonucleoprotein (RNP) complex with guideRNA (crRNA:tracrRNA). This combination reduces off-target effects to non-detectable levels and leads to Hifi hetero/homozygous knockin (*Vakulskas et al., 2018*). To avoid comprehensive and labor-intensive genotyping post-editing of clones (*Miura et al., 2018*), we took the advantage of fluorescent read-outs to monitor and enrich correct homology directed repair (HDR) by fluorescence-activated cell sorting (FACS) (*Li et al., 2017*; *Leonetti et al., 2016*). Pilot testing led to a high rate of successful knock-in human embryonic kidney (HEK)293T cells with expected localization of Rab11a (10–20%) or Rab11b (2–10%), respectively (*Figure 1—figure supplement 1F*). For further work, we selected a commonly used cell line model for ovarian cancer, A2780, and knocked-in mNeonGreen or mCherry after the start codon of Rab11a or Rab11b, respectively (*Figure 1D*). This integration, although less efficient (0.2–1%) compared to HEK293T, was specific as all sorted cells exhibited correct Rab11a/b localization, and our protocol was also successful in a second ovarian cancer cell line that has defective homologous recombination (BRCA-1 mutant COV362 *Beaufort et al., 2014*; *Figure 1E*). Western blotting with antibodies that recognize Rab11a, or both Rab11a and Rab11b, confirmed knock-in (*Figure 1F*). Consistent with the literature (*Li et al., 2017*; *Canaj et al., 2019*), a proportion of cells showed truncation of the linker flanking the FP (*Figure 1G*) but not in the integrated homologous arms, indicating that sorting cells for fluorescence (or another functional protein) is important for establishing in frame knock-in at the N-terminus preserving protein function. This potential caveat was demonstrated by our effort to knock-in an additional non-fluorescent component, antiGFPnanobody, together with mCherry. Although knock-in to the Rab11a locus showed the expected localization of mCherry-Rab11a in all sorted cells (*Figure 1—figure supplement 2A*), integration of the antiGFPnanobody sequence was not complete in ~90% of mCherry-positive cells (*Figure 1—figure supplement 2B-C*).

## DExCon supresses endogenous gene expression and re-activates in a doxycycline-dependent manner

Being able to visualize endogenous Rab11a/b, we analyzed the sequences of different promoters for their ability manipulate gene expression at the endogenous locus. The most promising candidate was the advanced non-leaky TRE3GS promoter of the third-generation tetracycline-inducible system (Tet-On 3G) as it offered conditional and reversible control regulated by the small molecule doxycycline (dox). Analysis of 365 bp TRE3GS promoter revealed four potential translation initiation

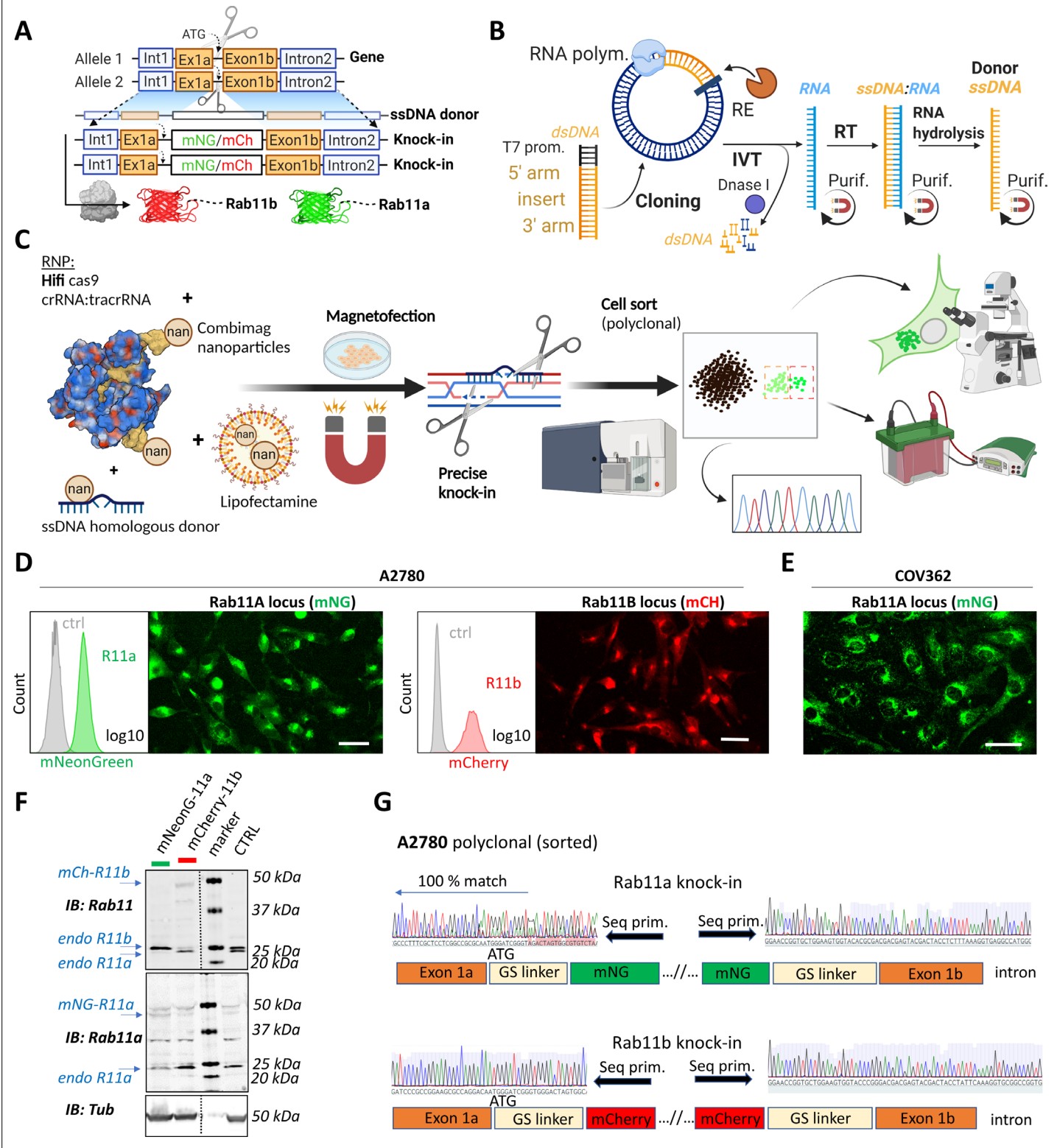

**Figure 1.** A rapid and efficient pipeline for CRISPR/Cas9 knock-in of fluorescent proteins at the endogenous locus of Rab11 family members. (**A–C**) Schematic representation of single-stranded DNA (ssDNA) mediated knock-in strategy of different fluorophores with corresponding homologous arms (A), the ssDNA preparation technique (IVT=in vitro transcription; RT=reverse transcription; RE=restriction endonuclease; purif.=purification using magnetic SPRI beads) (B), and our optimized pipeline (RNP=ribonucleoprotein complex) (C). Illustrations were created with BioRender.com. FACS and fluorescence widefield images (background subtracted) of mNeonGreen or mCherry knock-ins to Rab11a or Rab11b loci of (**D**) A2780 or (**E**) COV362

*Figure 1 continued on next page*

*Figure 1 continued*

ovarian cancer cell lines. Ctrl represent unmodified cells; scale bar=40 μm. (**F**) Immunoblots of mNeonGreen-Rab11a, mCherry-Rab11b, and unmodified (Ctrl) cells. Fluorescent antibodies, specific anti-Rab11a, or antibody targeting both Rab11a/b (Rab11) shown as black and white. Tubulin, loading control. (**G**) Chromatograms of mNeonGreen-Rab11a and mCherry-Rab11b cells with schematic of knock-in outcome (seq. prim=sequencing primer).

The online version of this article includes the following figure supplement(s) for figure 1:

**Figure supplement 1.** Rab11 family expression profiling.

**Figure supplement 2.** N-terminal knock-in of non-fluorescent coding elements.

ATG codons, all of which could be designed out of frame to the endogenous start codon, leading to multiple premature stop codons in transcripts driven by basal expression of the endogenous active promoter (*Figure 2A*). Tet-On 3 G transactivator was stably delivered to A2780 cells by lentivirus in tandem with mTagBFP (separated by the self-cleaving peptide T2A) and cells selected for mTagBFP fluorescence. The TRE3GS promoter sequence was introduced into the donor sequence upstream of mNeonGreen or mCherry and delivered by long ssDNA-mediated knock-in to the Rab11a or Rab11b locus, respectively, in A2780 cells. Sorted cells showed expression of corresponding FP-Rab11 with a perinuclear vesicle-enriched localization matching the endogenous protein, but only in the presence of dox, and decreased levels of non-modified endogenous Rab11a/b indicating that DExCon blocks expression at the endogenous locus (*Figure 2B and C*; *Figure 2—figure supplement 1A*).

To determine whether the DExCon system preserves the expression of splice variants of the N-terminally modified genes from the endogenous locus, we analyzed Rab11b splice variants in A2780 cells by quantitative PCR (qPCR) and semi-quantitative reverse transcription PCR (*Figure 2—figure supplement 1B*). We identified two Rab11b splice variants that could produce protein coding products of similar molecular weight to the single band observed in A2780 cells by western blotting (201 [24.5 kDa] and 202 [20 kDa]). The main Rab11b protein-coding splice variant 201 is ~4× more abundant than the protein-coding splice variant 202, consistent with eQTLs analysis of ovarian cancer (*Figure 1—figure supplement 1D*), and DExCon modification did not significantly influence their relative abundance (*Figure 2—figure supplement 1B*). Specific primers that detect splice variants 201, 202, and a third (short and likely untranslated) 204 variant indicated that the latter is expressed, but again the DExCon module did not change the ratio of Rab11b splice variants 201/202/204 upon dox induction. This suggests that the DExCon module can recapitulate the spectrum of splice variant expression from the Rab11b locus and provides an advantage over generation of a knock-out cell line and subsequent tet-inducible protein-coding rescue from a viral vector containing cDNA.

To exemplify the versatility of our CRISPR knock-in pipeline, we generated double mNeon-Green/mCherry knock-in at Rab11a/Rab11b loci, respectively, with and without the DExCon module (*Figure 2E and F*). In all combinations, endogenous Rab11a/Rab11b localization driven by dox-induced or endogenous promoter was significantly correlated, and both Rab11a and Rab11b concentrated in vesicles at the perinuclear recycling compartment (*Figure 2E and F*). By contrast, transient transfection of plasmids coding FP-Rab11a/Rab11b gave very bright overlapping signals, but large vesicles were observed particularly toward the cell periphery (*Figure 2G*). These findings were confirmed when red/green-FPs were switched (*Figure 2—figure supplement 2A, B*) and suggest that DExCon can induce expression from the endogenous gene locus that preserves physiological distribution of the protein within cells with signal intensities compatible with low light/exposure live microscopy (*Figure 2—figure supplement 2C,D*; *Videos 1–2*). This was further demonstrated by imaging Rab11a/b vesicles transporting the recognized cargo transferrin (TFN) (*Figure 2—figure supplement 2D*; *Video 3*) and by imaging cells in 3D environments. In 3D cell-derived matrix (CDM), higher exposures were required for visualization of conventional double knock-ins, and this led to a lag in capture between channels and significant bleaching of mCherry over time (*Figure 2—figure supplement 2F*; *Video 4*). However, DExCon knock-ins showed higher intensity and could therefore be visualized at lower exposure, avoiding caveats associated with lag (mis-localization) and photobleaching, revealing that while Rab11a vesicle intensity is consistent within polarized cells, Rab11b intensity is increased at the tips of protrusions relative to the region of the cell in front of the nucleus (*Figure 2—figure supplement 2E*).

We next explored the ability of DExCon to deliver full-length knock-in of GFP nanobody-mCherry sequence to Rab11 gene loci, as its flanking by an upstream TRE3GS sequence could improve the likelihood of complete integration and positive selection. Upon dox induction, all sorted cells exhibited

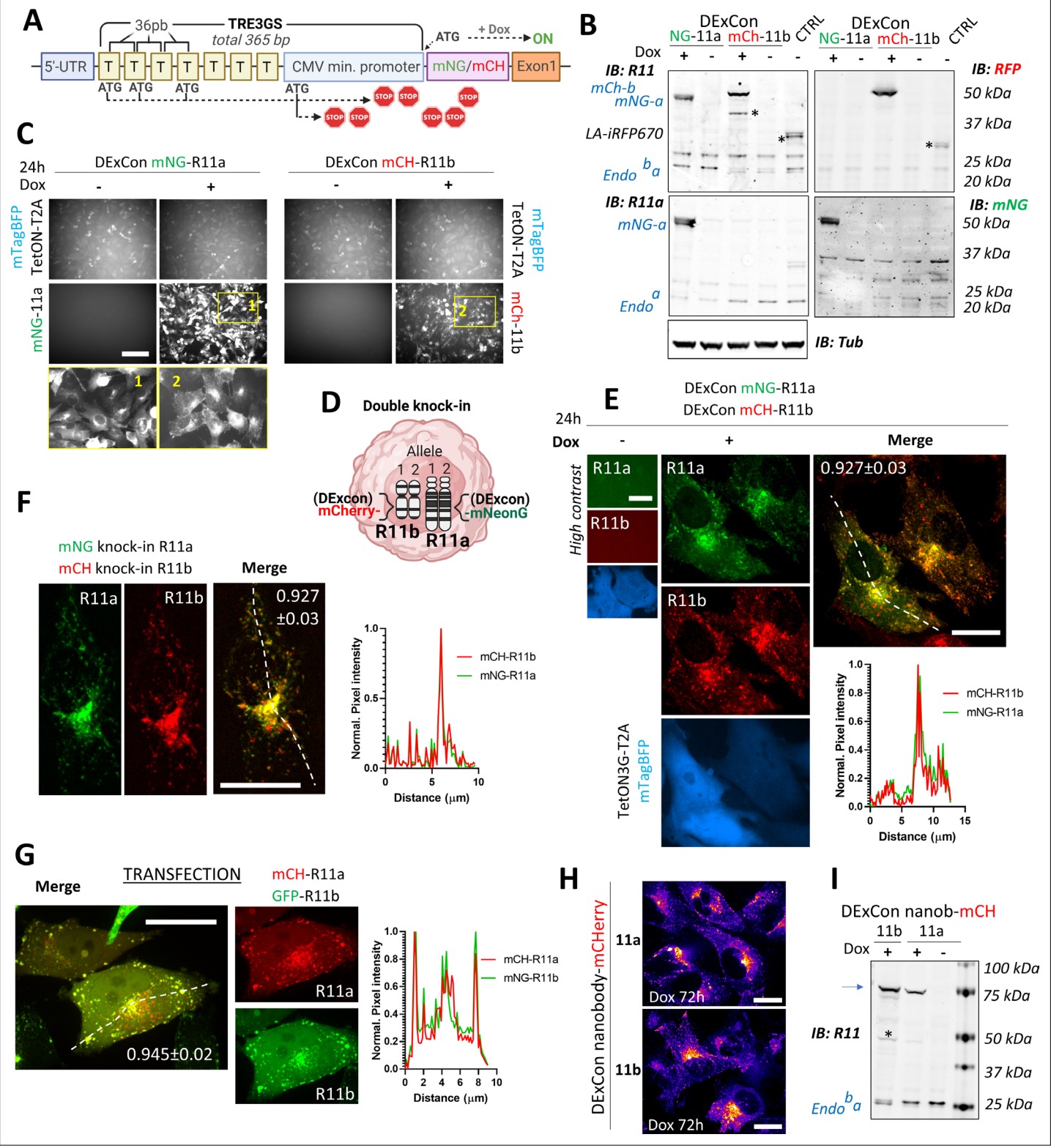

**Figure 2.** Reversible suppression of endogenous gene expression with DExCon (Doxycycline-mediated endogenous gene Expression Control). (**A**) DExCon schematic. (**B**) DExCon knock-in cells were doxycycline (dox) treated for 24–72 hr, sorted for mCherry or mNeonGreen fluorescence, and grown for 2 weeks without dox before re-analysis. Representative immunoblots of mNeonGreen-Rab11a or mCherry-Rab11b DExCon A2780 cells treated or not treated with dox for 48 hr (a=Rab11a; b=Rab11b) probed with anti-mNeonGreen (mNG), anti-mCherry (RFP), anti-Rab11a, or anti-Rab11 (targeting both Rab11a/b) antibodies shown as black and white. Tubulin (Tub), loading control. CTRL represents wt A2780 over-expressing Lifeact-

*Figure 2 continued on next page*

*Figure 2 continued*

iRFP670. Stars indicate mCherry/iRFP670 lower molecular weight band caused by fluorescent protein (FP) hydrolysis during sample preparation (***Gross et al., 2000***). For higher contrast, see ***Figure 2—figure supplement 1A***. (**C**) Live fluorescence images of mNeonGreen-Rab11a or mCherry-Rab11b DExCon (A2780) cells ±dox treatment (24 hr). Scale bar=100 µm. Area in yellow rectangle (1 or 2) is shown at the bottom with higher resolution. (**D**) Schematic of double knock-in, classical or DExCon, within same cell. (**E–G**) Spinning disk confocal images of (E) double DExCon Rab11a/b expressing Teton3G-T2A-mtagBFP±dox, (F) classical double mNeonGreen or mCherry knock-ins to Rab11a or Rab11b as indicated, and (G) A2780 co-transfected (magnetofection) with cDNA for GFP-Rab11b and mCherry-Rab11a. Maximum intensity Z-projection are shown, scale bar=20 µm. Plot profiles correspond to the dashed line and numbers reflect Pearson's cross-correlation coefficient (average ± SEM, n=14–22 cells). Spinning disk confocal images (**H**) or immunoblots (**I**) of antiGFPnanobody-mCherry DExCon cells (mCherry channel as 'gem' pseudocolor image Look-Up Table), modified in Rab11a or Rab11b locus treated ±dox for 72 hr. Scale bar=20 µm. Blue arrow points to the full-length fusion product.

The online version of this article includes the following figure supplement(s) for figure 2:

**Figure supplement 1.** DExCon (Doxycycline-mediated endogenous gene Expression Control) system allows re-expression of splicing isoforms from the endogenous locus.

**Figure supplement 2.** DExCon (Doxycycline-mediated endogenous gene Expression Control) controls Rab11a and Rab11b expression and preserves physiological localization.

the expected localization of Rab11a or Rab11b (***Figure 2H***) and full-length size on western blot (***Figure 2I***) indicating that encoding functionality at each of the 5' and 3' ends of ssDNA donor allows enrichment of accurately modified cells even for non-fluorescent coding genes.

## Re-activation of silenced genes by DExCon and LUXon (light responsive DExCon)

A2780 ovarian cancer cells show abundant Rab11a/b expression while the more restricted family member Rab25 is not expressed (***Figure 3A***). To study endogenous Rab25 in this cell line, we hypothesized that our DExCon strategy could re-activate Rab25 expression from the genome (***Figure 3B***). TRE3GS-mNeonGreen was delivered by long ssDNA-mediated knock-in to the Rab25 locus in A2780 cells expressing Tet-On 3G transactivator. Dox treatment led to mNeonGreen fluorescence with expected perinuclear localization of mNeonGreen-Rab25 in all sorted cells (in 2D and 3D-CDM; ***Figure 3C***) and specificity of integration was confirmed by western blot (***Figure 3—figure supplement 1A***) and sequencing (***Figure 3—figure supplement 1B***).

We next explored tunability of Rab25 expression. Exposing cells to different dox concentrations appeared to have a dose-dependent effect when analyzed by western blotting (***Figure 3—figure supplement 1C***); however, fluorescence microscopy revealed that the difference was primarily due

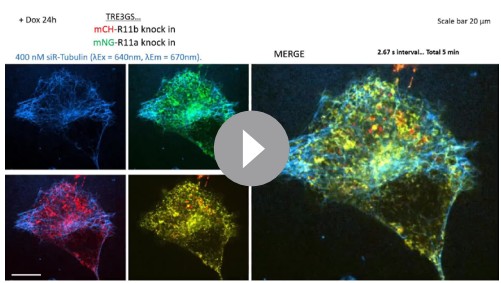

**Video 1.** Timelapse (Spinning disc 3i, 63×) of mNeonGreen-Rab11a/mCherry-Rab11b double DExCon (Doxycycline-mediated endogenous gene Expression Control) cells (A2780, dox for 24 hr) treated by siR-Tubulin (400 nM). Timelapse covers total 5 min with frame taken every 2.67 s (~10.3 s elapsed time per second of the movie); µ-Plate 96 Well Black (Ibidi cat.No 89626; #1.5 polymer coverslip, tissue culture treated; Opti-Klear Live Cell Imaging Buffer). Selected frame from this video is shown in Figure 2—figure supplement 2C.

https://elifesciences.org/articles/76651/figures#video1

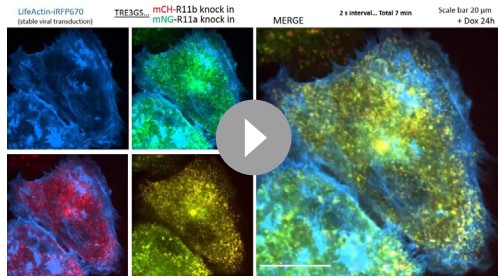

**Video 2.** Timelapse (Spinning disc 3i, 63×) of mNeonGreen-Rab11a/mCherry-Rab11b double DExCon (Doxycycline-mediated endogenous gene Expression Control) cells (A2780, dox for 24 hr) stably expressing LifeAct-iRFP670. Timelapse covers total 7 min with frame taken every 2 s (~14.5 s elapsed time per second of the movie); µ-Plate 96 Well Black (Ibidi cat.No 89626; #1.5 polymer coverslip, tissue culture treated; Opti-Klear Live Cell Imaging Buffer). Selected frame from this video is shown in Figure 2—figure supplement 2C.

https://elifesciences.org/articles/76651/figures#video2

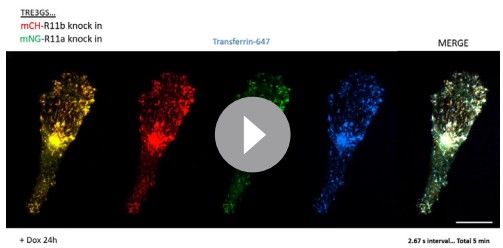

**Video 3.** Timelapse (Spinning disc 3i, 63×) of mNeonGreen-Rab11a/mCherry-Rab11b double DExCon (Doxycycline-mediated endogenous gene Expression Control) cells (A2780, doxycycline [dox] for 24 hr) recycling Alexa-647 labeled transferrin (25 µg/ml). Timelapse covers total 5 min with frame taken every 2.67 s (~16.2 s elapsed time per second of the movie); µ-Plate 96 Well Black (Ibidi cat.No 89626; #1.5 polymer coverslip, tissue culture treated; Opti-Klear Live Cell Imaging Buffer). Selected frame from this movie is shown in Figure 2—figure supplement 2D.

https://elifesciences.org/articles/76651/figures#video3

to the number of responding cells (*Figure 3D*). Nevertheless, we noticed a positive correlation between expression of Tet-On 3G transactivator, reported as mTagBFP fluorescence, and mNeonGreen-Rab25 (*Figure 3—figure supplement 1D*; *R*=0.43, p=0.0003). Taking advantage of this, we sorted for mNeonGreen-Rab25 expression and analyzed Rab25 and mTagBFP levels. Dox-induced Rab25 expression levels in 'low' sorted cells were comparable to endogenous Rab25 expression levels in OVCAR-3 ovarian cancer cell lines over 48 hr (*Figure 3E*) and Rab25 expression levels correlated with Tet-On 3G levels as expected (*Figure 3F*). The ability to re-induce previously sorted expression levels was preserved over several months under standard cell culture conditions and reached maximal fluorescence after 72 hr (*Figure 3E*). These results indicate that the DExCon approach was able to re-acti-vate endogenous silenced Rab25 on demand and levels could be tuned to match endogenous, physiologically relevant expression levels.

While DExCon was able to temporally control Rab25 expression, this method lacks spatial control. We therefore explored the newly developed PA-Tet-OFF/ON system (*Yamada et al., 2018*), which has not previously been applied at an endog-enous locus, to establish a light inducible 'LUXon' variant of DExCon. This system integrates the cryptochrome 2 cryptochrome-interacting basic helix-loop-helix 1 light-inducible binding switch with Tet-binding domains of Tet repressor (TetR) (residues 1–206) as the split DNA-binding domain, and the transcription activation domain of p65 (p65 AD) (*Figure 3G*). We tested PA-Tet-OFF or PA-Tet-ON and versions with improved blue-light sensitivity (ON2; OFF2), stably delivered by lentivirus. Cells stably expressing PA-Tet-ON2, PA-Tet-OFF1, or two were then edited by mNeonGreen DExCon knock-in to the Rab25 locus, illuminated by blue light for 24 hr and sorted, all showing the expected Rab25 endosomal localization (*Figure 3H*; *Figure 3—figure supplement 1E-G*). Cyclical exposure to light/dark led to repeatable re-silencing of Rab25, and the level of re-activation could be modulated by timing of blue light which is one of the major advantages of a light-inducible gene expression system (*Figure 3—figure supplement 1E-G*). As these systems responded to the weak blue-light exposure used (~6.8 W/m²), we found the addi-tional layer of dox control to prevent leakiness of the system very practical. Dox addition with blue light was necessary to re-activate Rab25 in PA-Tet-ON2 system (*Figure 3—figure supple-ment 1E*), while dox addition to PA-Tet-OFF1/2 completely blocked expression (*Figure 3—figure supplement 1F,G*). Interestingly, in addition to the expected Rab25 localization, we observed some Rab25 positive vesicles in close proximity of the nucleus. These vesicles were significantly less mobile compared to others suggesting that the nucleus could under some circumstances sterically block movement of vesicles (*Figure 3H*; *Videos 5 and 6*).

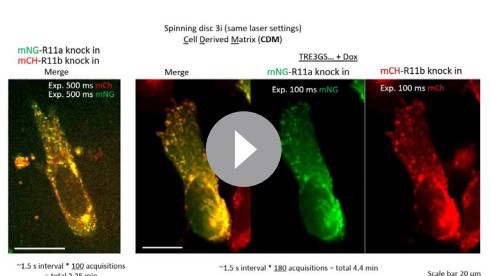

**Video 4.** Timelapse (Spinning disc 3i, 63×) of mNeonGreen-Rab11a/mCherry-Rab11b double knock-in A2780 cells±TRE3GS promoter (DExCon [Doxycycline-mediated endogenous gene Expression Control]; ±dox 24 hr) migrating in 3D cell-derived matrix. Timelapse covers total 2.25 min (left) or 4.4 min (right) with frame taken every ~1.5 s for both (~10.1 s elapsed time per second of the movie); µ-Plate 96 Well Black (Ibidi cat.No 89626; #1.5 polymer coverslip, tissue culture treated; Opti-Klear Live Cell Imaging Buffer). Selected frames from this movie are shown in Figure 2—figure supplement 2E,F.

https://elifesciences.org/articles/76651/figures#video4

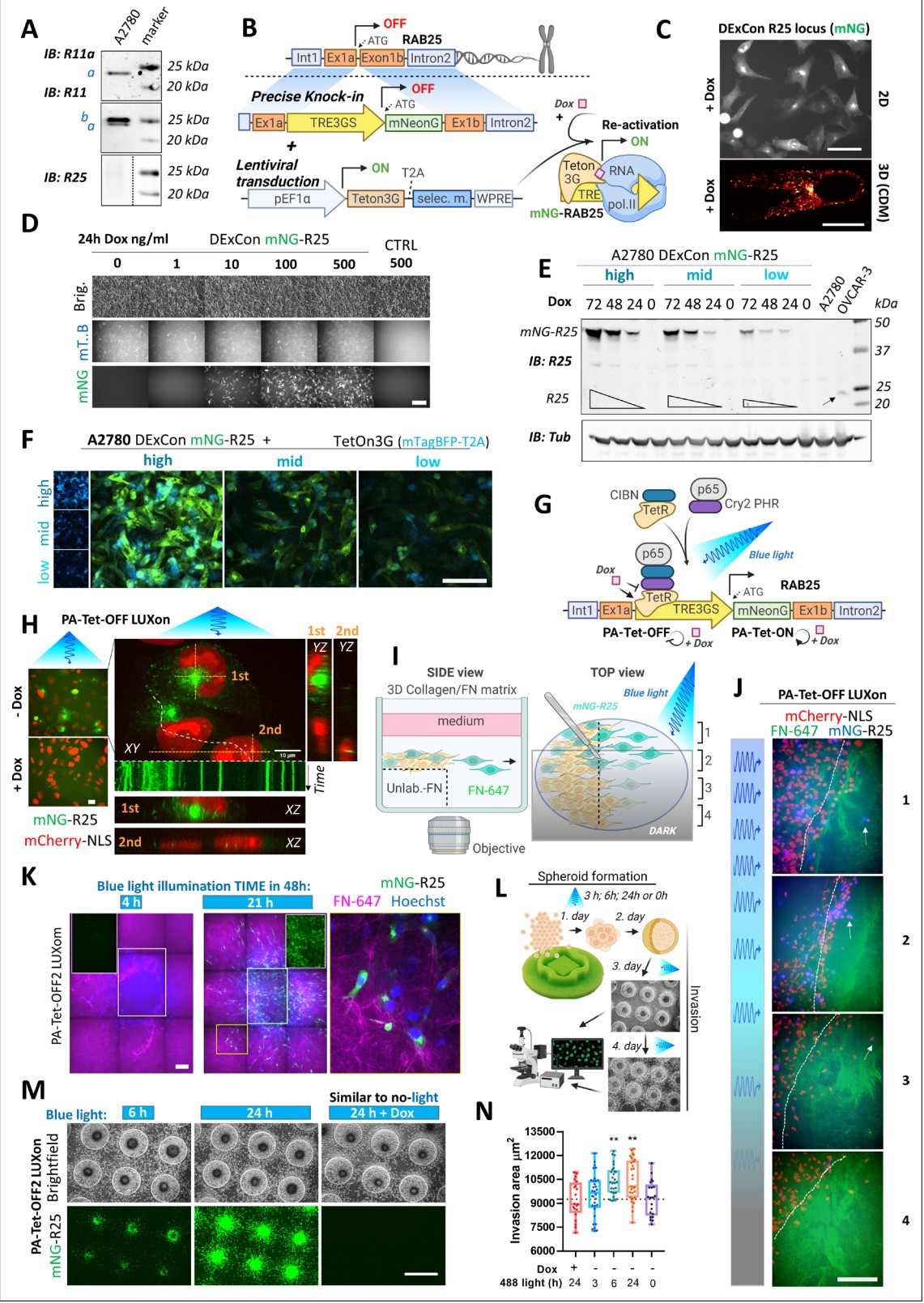

**Figure 3.** Spatiotemporal control of gene re-activation with DExCon (<u>D</u>oxycycline-mediated endogenous gene <u>Ex</u>pression <u>Con</u>trol) and LUXon (light responsive DExCon). (**A**) Immunoblots of A2780 cells lysates probed with antibodies specific for anti-Rab11a, targeting both Rab11a/b (Rab11) or Rab25 (a=Rab11a; b=Rab11b). Tubulin, loading control. (**B**) Schematic of the DExCon-mNeonGreen knock-in strategy and lentiviral transduction for doxycycline (dox)-dependent re-activation of Rab25 expression. (**C–D**) Rab25 DExCon knock-in cells were dox treated for 24–72 hr, sorted for

*Figure 3 continued on next page*

*Figure 3 continued*

mNeonGreen fluorescence followed by cell growth 2 weeks without dox before re-analysis. Live fluorescence images of re-activated Rab25 fused to mNeonGreen (mNG) in A2780 24 hr after dox treatment (250 ng/ml). (**C**) Top: Cells on tissue culture-treated plastic (scale bar=100 μm). Bottom: Cell in cell-derived matrix (CDM) (spinning disc confocal image; scale bar=20 μm). (**D**) Cells exposed to increasing dox concentration imaged by brightfield and fluorescence microscopy (mT.B=Teton3G-T2A-mTagBFP; brig=brightfield; Ctrl=unmodified cells). Scale bar=100 μm. (**E**) Immunoblots of mNeonGreen-Rab25 DExCon re-sorted cells as indicated in *Figure 3—figure supplement 1D* and re-induced with dox (200 ng/ml). Lysates were probed with antibodies specific for Rab25 or Tubulin. Black arrow indicates endogenous Rab25 in OVCAR-3 cells. (**F**) Spinning disc confocal images of mNeonGreen-Rab25 DExCon cells (Rab25, green; Teton3G, blue), re-sorted as indicated in *Figure 3—figure supplement 1D*. Scale bar=100 μm. (**G**) Schematic illustration of the photoactivatable split PA-Tet-OFF and PA-Tet-ON constructs (*Yamada et al., 2018*) combined with DExCon (LUXon). (**H**) Spinning disc confocal images (20 × [left] or 63 × [right] objectives) of mNeonGreen-Rab25 LUXon cells (Rab25 in green) expressing PA-Tet-OFF with mCherry-NLS reporter (red) 18 hr after being illuminated by blue light (10 hr) with or without dox treatment. Orthogonaere imaged live while recycll views of top (1) or two bottom cells (2) are also shown with kymograph corresponding to white dashed line (1 s interval, total 1 min). Scale bar: 10 μm. See *Videos 5–6*. (**I**) Schematic of 3D cell-zone exclusion invasion assay; 1–4 indicate zones with different light illumination intensities across same well, and these zones are also indicated in (**J**): mNeonGreen-Rab25 LUXon cells (Rab25 in blue) expressing PA-Tet-OFF with mCherry-NLS as nuclear reporter (red) migrated for 24 hr into cell-free collagen matrix labeled by fibronectin (FN-647 shown as green) while being illuminated by blue light of varying intensity. Dashed line indicates scratch and white arrows indicate the most invasive cells. Scale bar=100 μm. See *Video 7*. (**K**) Spinning disc confocal images of spheroids formed by mNeonGreen-Rab25 LUXon (PA-Tet-OFF2) cells illuminated with blue light for different times across 2 days as indicated (total 4 hr vs 21 hr). Cells invading collagen matrix supplemented with FN-647 (magenta) were labeled with Hoechst 3342 (blue) for 1 hr prior imaging. Merge of all three channels, white rectangles for the mNeonGreen (Rab25, green) channel only, or zoom of the yellow rectangle is shown. Scale bar=100 μm. (**L**) Schematic of spheroid invasion assay 'on chip' and illumination protocol used in (**M**). mNeonGreen-Rab25 LUXon (PA-Tet-OFF2) cells (scale bar=1 mm) together with (**N**) quantification (n=20–32 from three independent experiments, one-way ANOVA Tukey post hoc test). All schematic illustrations were created with BioRender.com.

The online version of this article includes the following figure supplement(s) for figure 3:

**Figure supplement 1.** Re-activation of Rab25 expression.

To demonstrate local induction of gene expression from the endogenous locus, we utilized a modified 3D cell-zone exclusion assay (*Gemperle et al., 2019*). In this assay, a monolayer of the cells grown on a thick layer of collagen supplemented with non-labeled fibronectin is wounded and then overlaid with another layer of collagen supplemented with far-red labeled fibronectin (*Figure 3I*). The advantage of this approach is that system can be partly illuminated and Rab25 expressing cells can be monitored migrating toward the far-red fibronectin labeled cell-free area. This approach allowed us to demonstrate that Rab25 expression can be induced locally in 3D matrices and is suggestive of a link between Rab25 re-expression levels and increased invasiveness (*Figure 3J*; *Video 7*). To quantify this, we next illuminated cell spheroids overlaid

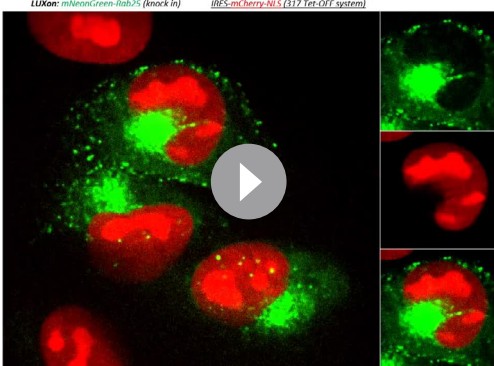

**Video 5.** Timelapse (Spinning disc 3i, 63×) of mNeonGreen-Rab25 LUXon (light responsive DExCon) cells (Rab25 as green) expressing PA-Tet-OFF with mCherry-NLS reporter (red) 18 hr after being illuminated by blue light (10 hr). Timelapse covers total 1 min with frame taken every 1 s (~6.7 s elapsed time per second of the movie); μ-Plate 96 Well Black (Ibidi cat.No 89626; #1.5 polymer coverslip, tissue culture treated; Opti-Klear Live Cell Imaging Buffer). Selected frame from this video is shown in Figure 3H.

https://elifesciences.org/articles/76651/figures#video5

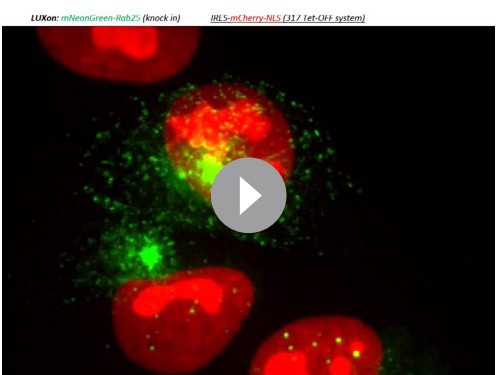

**Video 6.** 3D projection (Spinning disc 3i, 63×) of mNeonGreen-Rab25 LUXon (light responsive DExCon) cells (Rab25 as green) expressing PA-Tet-OFF with mCherry-NLS reporter (red) 18 hr after being illuminated by blue light (10 hr). μ-Plate 96 Well Black (Ibidi cat.No 89626; #1.5 polymer coverslip, tissue culture treated). Selected frame from this movie is shown in Figure 3H.

https://elifesciences.org/articles/76651/figures#video6

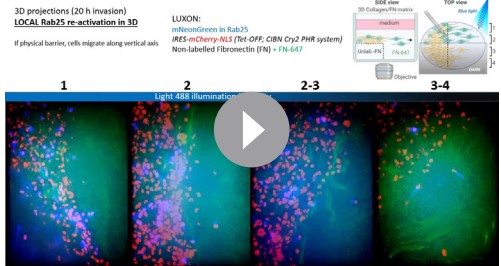

**Video 7.** 3D projection (Spinning disc 3i, 63×) of mNeonGreen-Rab25 LUXon (light responsive DExCon) cells (Rab25 as blue) expressing PA-Tet-OFF with mCherry-NLS reporter (red) 20 hr after being spatiotemporally illuminated by blue light (10 hr). Cells are invading to cell-free collagen matrix labeled by FN-647 (green) while being illuminated by blue light of varying intensity (see experimental set-up, right corner). µ-Plate 96 Well Black (Ibidi cat.No 89626; #1.5 polymer coverslip, tissue culture treated) partly covered by black plasticine. Selected frame from this movie is shown in Figure 3J.

https://elifesciences.org/articles/76651/figures#video7

by 3D matrix, modulating the exposure to blue light (*Figure 3K–N*). Longer exposure to blue light led to brighter mNeonGreen signal in both PA-Tet-OFF/ON systems with expected Rab25 localization in polarized/elongated cells interacting with matrix fibrils (*Figure 3K*; *Figure 3—figure supplement 1H*). When all cells were pre-illuminated 2 days before invasion to induce Rab25 expression, we saw a significant dose-dependent effect of Rab25 induction via light exposure for over 6 hr on invasion that was not observed in the presence of dox in the TET-OFF system (*Figure 3L–N*). Adjusting the illumination protocol (*Figure 3—figure supplement 1I*), activating Rab25 expression with constant blue light for 24 hr before spheroid formation/invasion, led to a light-dose dependent effect where the most invasive cells were those illuminated the longest approach (*Figure 3—figure supplement 1J,K*). This demonstrates that tight control of gene expression levels and their impact on complex cell behavior can be easily achieved/studied by the LUXon system.

## Triple knock-in determines Rab11 family colocalization

After generating mNeonGreen-Rab11a/mCherry-Rab11b double knock-in cells, we challenged ourselves to generate triple knock-in A2780 cells (Rab11a/Rab11b/Rab25; *Figure 4A*). To do so, mNeonGreen-Rab11a/mCherry-Rab11b double knock-in cells were further genetically manipulated to introduce mTagBFP2 as part of the DExCon module controlling Rab25 expression. Dox treatment successfully specifically re-activated Rab25, visible as mTagBFP2, on the background of double knock-in mNeonGreen-Rab11a/mCherry-Rab11b cells (*Figure 4—figure supplement 1A,B*). Interestingly, activation of Rab25 expression increased the localization of Rab11a and Rab11b at ERC spots, suggesting that Rab11a/b/25 networks are interconnected and can be modulated by Rab25 expression (*Figure 4—figure supplement 1C*), consistent with redistribution of Rab11a in Madin-Darby canine kidney (MDCK) cells upon the introduction of Rab25 expression (*Casanova et al., 1999*). Because DExCon-mTagBFP2-Rab25 fluorescence intensity was less intense than that of DExCon-mNeonGreen-Rab25 and Rab11a/b knock-ins, triple knock-in cells were re-sorted for bright mTagBFP2 signal and dox treatment kept over 94 hr prior fluorescence microscopy highlighting the tunability of expression with DExCon (*Figure 4—figure supplement 1A,B*).

Rab11 family members have been reported to colocalize with TFN receptor and control its recycling (*Ullrich et al., 1996*; *Schlierf et al., 2000*). To compare the colocalization of Rab11 family members and their potential contribution to the TFN receptor recycling, we allowed cells to internalize Alexa-647 labeled TFN and analyzed colocalization using AiryScan confocal and lattice light sheet microscopy (*Figure 4B–F*; *Figure 4—figure supplement 1B-E*; *Videos 8–12*). While there was clear colocalization between Rab11a/b/25 and TFN, some vesicles were highly enriched for Rab25 compared to Rab11a/b but lacked TFN. Rab25 enriched vesicles negative for TFN were found in close proximity to the nucleus in ~90% of triple knock-in cells (*Figure 4B*), both in 2D and 3D environments (*Figure 4C*; *Videos 9–12*). To quantify the colocalization of vesicles, we tracked ~46,000 vesicles per Rab11 family member. This analysis revealed that there is no significant difference in TFN colocalization with Rab11 family members, but Rab11a colocalizes with Rab11b to a significantly higher extent than with Rab25, or Rab11b with Rab25 (*Figure 4D*). Rab11a/b/25 largely colocalize with TFN, one another, or all at the same time (16.1% of all Rab11 vesicles/26.5% of TFN vesicles; *Figure 4E*) and that differences between Rab11 family members are mainly due to the unique Rab25 recruitment to a subset of vesicles (*Figure 4F*; *Figure 4—figure supplement 1D,E*).

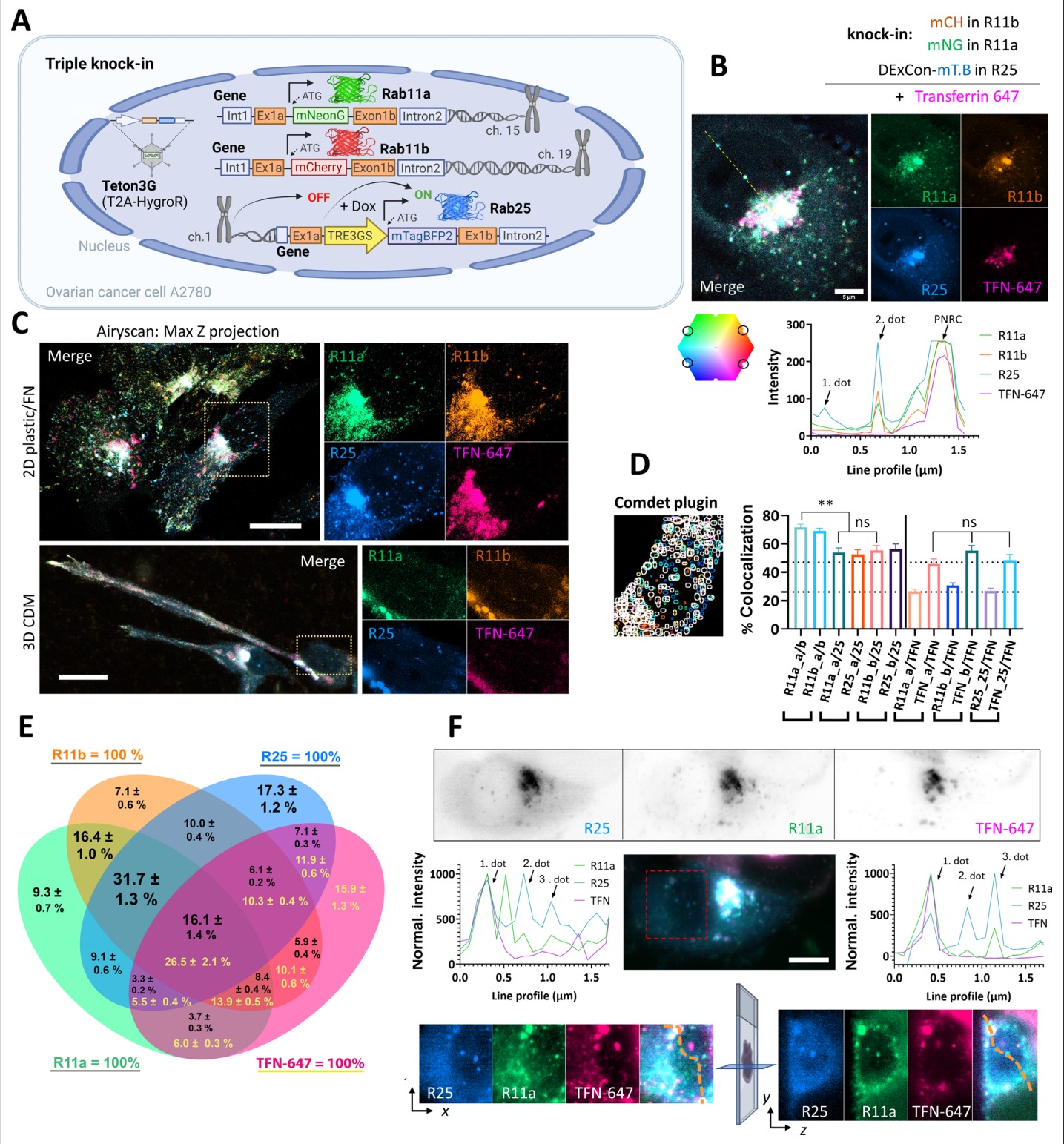

**Figure 4.** Simultaneous visualization of Rab11 family members. (**A**) Schematic of triple knock-in A2780 cells (mNeonGreen-Rab11a, mCherry-Rab11b, DExCon mTagBFP2-Rab25) created with BioRender.com. (**B–D**) mNeonGreen-Rab11a/mCherry-Rab11b knock-in A2708 cells were further modified with a DExCon-mTag-BFP2 module at the Rab25 locus (Tet-On transactivator introduced by lentivirus with hygromycin selection). Airyscan confocal fluorescence images of triple knock-in cells treated by doxycycline (dox) (>94 hr) trafficking Alexa-647 labeled transferrin (TFN-647, 15–60 min). Colors represent Rab11s as indicated and line profile corresponds to yellow dashed line. (**B**) Scale bar=5µm. (**C**) Maximum intensity Z-projections: top (2D, FN-coated), bottom (3D cell-derived matrix [CDM]). Scale bar=20µm; see also *Videos 9–10*. (**D–E**) dox induced (>94 hr) triple knock-in A2780 cells recycling

*Figure 4 continued on next page*

*Figure 4 continued*

TFN-647 were imaged and mNeonGreen/mCherry/mTagBFP2/TFN-647 positive vesicles tracked using Comdet plugin. (**D**) Bar graph representing the percentage of colocalizing vesicles (100 cells, 46,000 vesicles [Rab11s]; 23,000 vesicles [TFN]) and the contribution of individual channels (25- Rab25; TFN- transferrin; a- Rab11a; b- Rab11b). One-way ANOVA Tukey post hoc test used for statistical analysis. (**E**) Venn diagram with the percentage of colocalization for every channel as 100% total; black numbers for Rab11s; yellow for TFN-647. (**F**) Lattice light-sheet imaging of dox induced (>94 hr) triple knock-in A2780 cells recycling TFN-647 are shown in gray as individual channels (one focal plane) or as maximum intensity Z-projection for merged channels; scale bar=10 μm. Line profiles (normalized to maximal intensity, background subtracted) correspond to orange dashed lines, which connects the same vesicles (1. and 3. dot) in the 3D cell volume. See also *Figure 4—figure supplement 1D*,E and *Videos 11 and 12*.

The online version of this article includes the following figure supplement(s) for figure 4:

**Figure supplement 1.** mNeonGreen-Rab11a, mCherry-Rab11b, and DExCon-mTagBFP2-Rab25 triple knock-in.

These results demonstrate that DExCon is a tool well suited for microscopy that allows multiple gene modifications simultaneously, including re-activation of previously silenced genes. Furthermore, complete integration can be verified using fluorescence-based sorting in polyclonal populations, and the approach provides a relatively simple platform for conditional rescues/knock-outs that are, in contrast to FLP/Cre-driven recombination, reversible.

## Protein expression kinetics, stability, and dynamics revealed using DExCon

Intrigued by slow expression kinetics of re-activated mNeonGreen-Rab25 DExCon in A2780 cells, we generated Rab11a and Rab11b DExCon cells to compare kinetics. Western blot suggested that we can detect mNeonGreen-Rab11a from 4 hr and mCherry-Rab11b from 6 hr after dox induction, while Rab25 DExCon was not detected before 8 hr (*Figure 5A*). The rate of dox induced expression was then quantified on individual DExCon Rab11 family gene modifications. mNeonGreen-Rab11a DExCon reached ~75% of its maximal expression peak at 24 hr, significantly faster than other family members: at 24 hr mCherry-Rab11b DExCon reached ~50% peak expression, whereas mNeonGreen-Rab25 DExCon only reached ~40% (*Figure 5B*). Similar results were also obtained from nanobody-mCherry-DExCon modifications (*Figure 5—figure supplement 1A,B*), suggesting that observed differences can be explained by differences in the regulation of Rab11 gene expression and not by different FPs.

Since Rab25 is silenced in A2780, we performed bioinformatic analysis of its methylation status. We found that there is an inverse relationship between methylation of the Rab25 promoter and its expression, where decreased methylation is correlated with increased gene expression, at both mRNA and protein levels across all cancer types (*Figure 5C*; *Figure 5—figure supplement 1C*). ATAC-seq dataset analysis further suggested that the chromatin structure of the silenced Rab25 locus, both in promoter and enhancer sequences, is less accessible for transcriptional machinery to drive

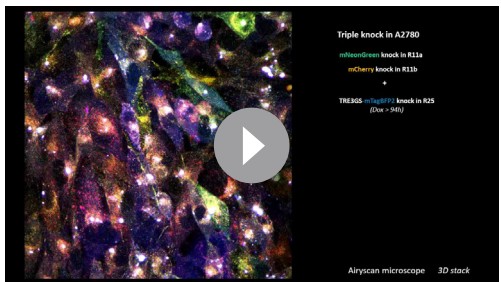

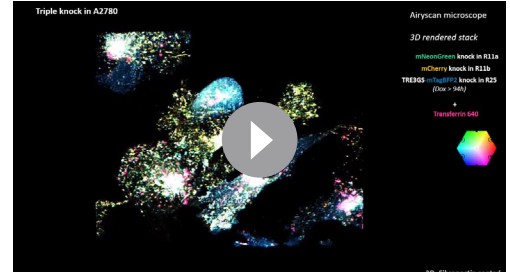

**Video 8.** 3D projection of triple knock-in A2780 cells (mNeonGreen-Rab11a, mCherry-Rab11b, mTagBFP2-Rab25 DExCon) treated by doxycycline (dox) (>94 hr). μ-Plate 96 Well Black (Ibidi cat.No 89626; #1.5 polymer coverslip, tissue culture treated; Opti-Klear Live Cell Imaging Buffer). AiryScan LSM880 (63×). Selected frames from this movie are shown in Figure 4—figure supplement 1B.

https://elifesciences.org/articles/76651/figures#video8

**Video 9.** 3D rendered model (ZEN black software) of triple knock-in A2780 cells (mNeonGreen-Rab11a, mCherry-Rab11b, mTagBFP2-Rab25 DExCon) treated by doxycycline (dox) (>94 hr). Cells were imaged live while recycling Alexa-647 labeled transferrin (30 min) on glass-bottom dish (MatTek, Ashland, MA, USA; Opti-Klear Live Cell Imaging Buffer) coated with 10 μg/ml fibronectin (FN) using AiryScan LSM880 (63×). Selected frames from this movie, raw un-rendered, are shown in Figure 4C.

https://elifesciences.org/articles/76651/figures#video9

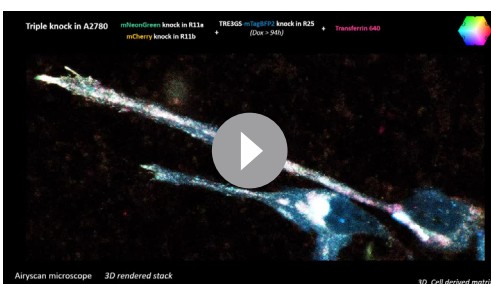

**Video 10.** 3D projection of triple knock-in A2780 cells (mNeonGreen-Rab11a, mCherry-Rab11b, mTagBFP2-Rab25 DExCon) treated by doxycycline (dox) (>94 hr). Cells were imaged live while recycling Alexa-647 labeled transferrin (30 min) and migrating in Cell Derived Matrix (3D). AiryScan LSM880 (63x); Glass-bottom dish (MatTek, Ashland, MA, USA); Opti-Klear Live Cell Imaging Buffer. Selected frames from this movie are shown in Figure 4C.

https://elifesciences.org/articles/76651/figures#video10

**Video 12.** Triple knock-in A2780 cells (mNeonGreen-Rab11a; mCherry-Rab11b; mTagBFP2-Rab25 DExCon) treated by doxycycline (dox) (>94 hr) with Alexa-647 labeled transferrin recycled for 30 min. Cells were imaged fixed on FN-coated 5 mm coverslip. 3D projection animation generated using Imaris Cell Imaging Software. Opti-Klear Live Cell Imaging Buffer; 3i Lattice LightSheet microscope. Selected frames from this animation are shown in Figure 4—figure supplement 1E.

https://elifesciences.org/articles/76651/figures#video12

transcription (*Figure 5—figure supplement 1C*). To prove that slower Rab25 DExCon expression kinetics in A2780 cells can be simply explained by the decrease of DNA accessibility for transcription, we knocked in TRE3GS-mNeonGreen to the Rab11a or Rab25 loci of OVCAR-3 cells, an ovarian cancer line where both Rab11a and Rab25 are endogenously expressed. Dox induction again led to the correct localization of Rab11a or Rab25, as expected (*Figure 5—figure supplement 1D*), but this time there was no delay in the normalized expression kinetics of Rab25, as was observed for A2780 (*Figure 5D*). Interestingly, in both A2780 and OVCAR-3 cells Rab11a DExCon always led to greater fluorescence intensity compared to Rab25 DExCon, suggesting additional mechanisms of protein level control (*Figure 5—figure supplement 1E*).

We further explored the possibility of simultaneous visualization of different alleles of the same gene (*Figure 5E*; *Figure 5—figure supplement 1F*, *Figure 5—figure supplement 2A-C*). We were able to successfully modify a second allele of mCherry-Rab11a, mCherry-Rab11b, or DExCon-mCherry-Rab11b modified cells by knocking in DExCon-mNeonGreen modules (*Figure 5F*; *Figure 5—figure supplement Figure 5—figure supplements 1F and 2A-C*). Dox-mediated induction led to consistent almost identical localization in all cells with alleles modified with mCherry and DExCon-mNeonGreen for both Rab11a and Rab11b (*Figure 5—figure supplement 2A-C*). Consistent with previous results (*Figure 2E*; *Figure 2—figure supplement 2E*), DExCon-mNeonGreen-Rab11a was slightly more diffuse than DExCon-mCherry-Rab11b in the same cells (*Figure 5—figure supplement 2D*). This demonstrates that DExCon-induced Rab11 expression levels do not appreciably alter localization of the GTPases and generate signals compatible with routine imaging.

Unexpectedly, expression kinetics of double DExCon-mCherry/DExCon-mNeonGreen Rab11b cells were noticeably different between mCherry and mNeonGreen modified Rab11b (*Figure 5H*; *Video 13*). While the mNeonGreen kinetics of Rab11a expression were still slightly faster than mNeonGreen of Rab11b, the mCherry Rab11b fluorescence displayed significant delay similar in both Rab11a/b and Rab11b double DExCons

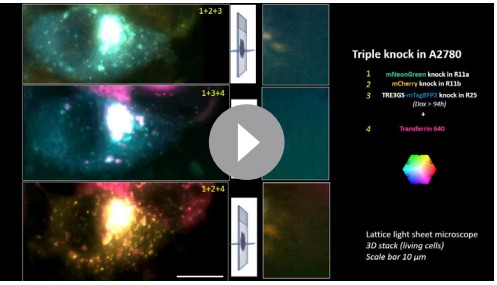

**Video 11.** Triple knock-in A2780 cells (mNeonGreen-Rab11a; mCherry-Rab11b; mTagBFP2-Rab25 DExCon) treated by doxycycline (dox) (>94 hr). Cells were imaged live while recycling Alexa-647 labeled transferrin (30–60 min) on FN-coated 5 mm coverslip. 3D projections and optical sections are shown. Opti-Klear Live Cell Imaging Buffer; 3i Lattice LightSheet microscope. Selected frames from this movie are shown in Figure 4F and Figure 4—figure supplement 1D.

https://elifesciences.org/articles/76651/figures#video11

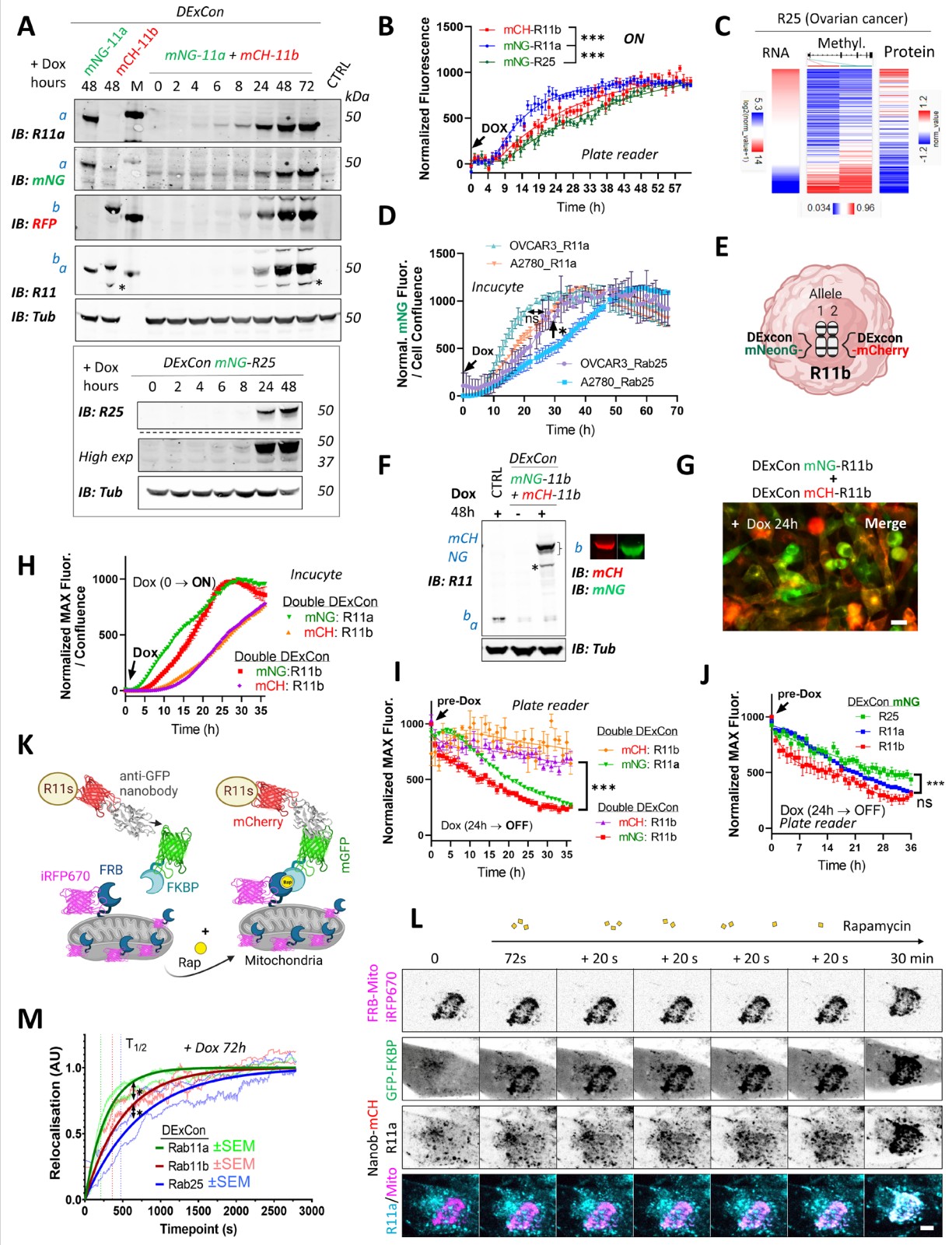

**Figure 5.** DExCon (<u>D</u>oxycycline-mediated <u>e</u>ndogenous gene <u>Ex</u>pression <u>Con</u>trol) reveals protein expression kinetics and dynamics of relocalization. (**A**) Immunoblots of <u>mNeonG</u>reen-Rab11a and <u>mCh</u>erry-Rab11b DExCon or double DExCon A2780 cells (top) and mNeonGreen-Rab25 DExCon cells (bottom) treated or not treated with doxycycline (dox) for the time indicated (hours). Anti-mNeonGreen (mNG), anti-mCherry (RFP; mCH), anti-Rab25, anti-Rab11a, or anti-Rab11 targeting both Rab11a/b (Rab11) were used to probe expression levels. Tubulin (Tub), loading control. M=marker;

*Figure 5 continued on next page*

*Figure 5 continued*

CTRL=unmodified wt A2780; a=Rab11a, b=Rab11b, c=Rab25±fluorophore knock-in. Stars indicate mCherry/iRFP670 lower molecular weight band caused by hydrolysis during sample preparation *Gross et al., 2000*. (**B**) Comparison of expression kinetics of DExCon-modified mNeonGreen-Rab11a/mCherry-Rab11b/mNeonGreen-Rab25 (dox added at t=0, arrow) using a BioTek Synergy H1 microplate reader. Cells were seeded confluent and fluorescence normalized to the maximal intensity. One-way ANOVA and Holm-Sidak post hoc tests were used for statistical analysis: n=6–9 from three independent experiments comparing normalized fluorescence intensity at 24 hr. (**C**) Bioinformatic analysis of DNA promoter methylation (Methylation27K) of Rab25 and its expression (RNAseq; RPPA) in ovarian cancer (OV) using UCSC Xena (TCGA Pan-Cancer study). (**D**) Comparison of expression kinetics of DExCon modified mNeonGreen-Rab11a or mNeonGreen-Rab25, generated in A2780 or OVCAR-3 cells, induced by dox at t=0 (left arrow) and analyzed using Incucyte S3 imaging system. Readings were normalized to maximal fluorescence intensity and cell confluence (representative graph is shown with mean ± SD; one-way ANOVA and Tukey post hoc test used for statistical analysis). (**E**) Schematic of double DExCon modification, targeting both Rab11b alleles/loci with different fluorophores within the same cell. (**F**) Immunoblots or (**G**) fluorescence images (scale bar=100 μm) of mNeonGreen/mCherry-Rab11b double DExCon cells A2780 treated±dox. Blots were probed with anti-mNeonGreen (mNG), anti-mCherry (RFP), and anti-Rab11 targeting both Rab11a/b (Rab11)-specific antibodies. Tubulin (Tub), loading control. CTRL=unmodified wt A2780; a=Rab11a; b=Rab11b±knock-in fluorophore. (**H–I**) Comparison of mNeonGreen-Rab11a/mCherry-Rab11b or mNeonGreen-Rab11b/mCherry-Rab11b double DExCon expression kinetics (**H**) triggered by dox (arrow; normalized and shown as in D) or their protein stability (**I**). (**J**) Comparison of DExCon induced mNeonGreen-Rab11a/Rab11b/Rab25 protein stability. (**I–J**) Cells pre-induced with dox for 24 hr followed by dox removal (see arrow; microplate reader BioTek Synergy H1). Cells were seeded confluent and fluorescence normalized to the maximal intensity. Mean ± SEM (9–18 repeats gathered from three to six independent experiments). Curves were fitted by fifth-order polynomial function; one-way ANOVA with Tukey post hoc test used for statistical analysis. (**K**) Schematic of knocksideways combined with DExCon antiGFPnanobody-mCherry-Rab11 family modification. (**L**) Representative confocal fluorescence timelapse images of antiGFPnanobody-mCherry-R11a DExCon modified cells induced with dox for 72 hr (scale bar=5 μm; see *Figure 5—figure supplement 3A*,B for Rab11b and Rab25). Cells were co-magnetofected with FRB-Mito-iRFP670 and GFP-FKBP on 96 well ibidi imaging plate 24 hr before imaging and their heterodimerization induced by 200 nM rapamycin as indicated. Halftime of Rab11a/b/25 relocalization fitted and quantified (**M**); for statistics see *Figure 5—figure supplement 3C*, n=10–18 from three to four independent experiments.

The online version of this article includes the following figure supplement(s) for figure 5:

**Figure supplement 1.** Protein expression kinetics and stability are revealed using DExCon (Doxycycline-mediated endogenous gene Expression Control).

**Figure supplement 2.** Simultaneous visualization of different alleles of the same Rab11 gene by using DExCon (Doxycycline-mediated endogenous gene Expression Control).

**Figure supplement 3.** Protein dynamics are revealed using DExCon (Doxycycline-mediated endogenous gene Expression Control).

cells. When DExCon cells were pre-induced with dox for 24 hr, followed by dox removal, a decrease in fluorescence was observed over time (*Figure 5I*). mCherry-Rab11b showed remarkably stable fluorescence, whereas mNeonGreen-Rab11b was lost more rapidly. Stable fluorescence of mCherry was also observed for nanobody-mCherry DExCon Rab11s (*Figure 5—figure supplement 1G*). By contrast, mNeonGreen fluorescence continuously declined overtime, where Rab11a and Rab11b decreased in intensity more rapidly than Rab25 following dox removal (*Figure 5J*). This suggests that the FP is an important consideration when using FP-knock-in to understand protein dynamics but opens the possibility of modifying individual (e.g. mutant vs non-mutant) alleles within cells to study their stability and function.

The localization of Rab GTPases is fundamental to their function, but the relative stability of Rab11 family members at their endogenous location is not known. We therefore combined DExCon with knock-sideways, a method of protein relocalization within live cells (*Robinson et al., 2010*), to give insight into the dynamics of Rab11a, Rab11b, and Rab25 localization. Relocalization to mitochondria was achieved by rapamycin (rap) induced dimerization of FKBP-mGFP with mitochondrially targeted mito-FRB-iRFP670 in GFPnanobody-mCherry DExCon modified A2780 cells (*Figure 5K*) and the dynamics of Rab11 family members analyzed by time-lapse imaging (*Figure 5L*; *Figure 5—figure supplement 3A,B*).

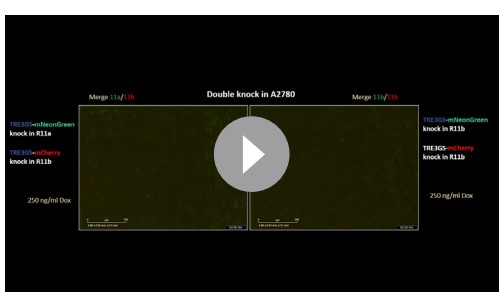

**Video 13.** Timelapse (Incucyte S3 system, 20x) of protein expression kinetics of mNeonGreen-Rab11a/mCherry-Rab11b or mNeonGreen-Rab11b/mCherry-Rab11b double DExCon cells (A2780) treated by doxycycline (dox) (from time 0). Timelapse covers total 49 hr with frame taken every 30 min (~30 min elapsed time per second of the movie); 96 well tissue culture plates (Corning); RPMI fenol-free media. Selected frames from this movie are shown in Figure 5G.
https://elifesciences.org/articles/76651/figures#video13

Direct comparison of Rab11a/b/25 dynamics revealed an ultra-fast relocalization of Rab11a ($t_{1/2}$=209 s), significantly slower Rab11b relocalization ($t_{1/2}$=362 s), and slower still dynamics of Rab25 ($t_{1/2}$=476 s) (*Figure 5M*; *Figure 5—figure supplement 3C*).

Taken together, these results show our DExCon approach can be used to reveal subtle but significant differences in protein expression kinetics and protein dynamics, and that despite their similarity at the protein level, such differences are observed within the Rab11 family.

## DExogron: a tunable approach to modify protein levels within cells

Because removal of dox did not rapidly decrease DExCon-driven protein levels, we added a second layer of control utilizing the improved AID system reported with low-basal degradation (*Li et al., 2019*). We combined our DExCon module with an auxin-inducible destabilizing domain (short degron miniIAA7) that allows proteasomal degradation in an auxin-dependent manner to generate DExogron gene modifications (*Figure 6A*).

Because we were able to independently target both alleles of Rab11a/b, we first combined our knock-in strategy with knock-in of a cassette composed of puromycin (PuroR) or blasticidine (BlaR) resistance markers and PolyA signal to block endogenous expression (*Figure 6B*; *Figure 6—figure supplement 1A*). In A2780 cells, inclusion of an additional promoter sequence (SFFV) was required because endogenous promoter-driven expression was insufficient to yield antibiotic resistance, in contrast to HEK293T cells. Sequential targeting of the second Rab11a or Rab11b allele in DExCon cells using PuroR or BlaR, significantly improved knock-in outcome both in DExogron, DExCon Rab11b, or double DExCon Rab11a/b cells (*Figure 6—figure supplement 1B-D*) and thus successfully generated near-complete knock-in/knock-out cells with only one allele of Ra11a or b under dox control (rescue). Notably, simultaneous targeting of two Rab11a alleles by DExCon and PuroR dramatically increased the proportion of DExCon knock-in cells surviving PuroR treatment, from 0.2 to 25% (*Figure 6—figure supplement 1E*). We successfully generated combinations of DExCon/DExogron-Rab11a/b, with additional PuroR/BlaR knock-ins (*Figure 6C*; *Figure 6—figure supplement 1F*). Dox treatment led to rescue of protein expression as expected and simultaneous addition of Indole-3-acetic acid (IAA as a source of auxin) efficiently depleted dox-induced expression only when the degron tag was present (DExogron miniIAA7-mCherry-Rab11b) (*Figure 6C*; *Figure 6—figure supplement 1D, F*; *Videos 14 and 15*). Flow cytometry analysis confirmed that IAA/dox co-treatment (24–48 hr) decreases DExogron-mCherry-Rab11b fluorescence to the level of un-treated Rab11b miniIAA7-mCherry knock-in cells with degradation half time kinetics of 1.7±0.1 hr (*Figure 6D and E*; *Figure 6—figure supplement 1G*; *Video 14*).

Introducing a degron tag using one donor without additional DExCon modification led to a decrease in mCherry fluorescence (*Figure 6D*) and revealed two additional drawbacks. A significant proportion of cells did not respond to IAA likely due to the truncation of the miniIAA7 sequence during HDR (*Figure 6—figure supplement 1H*). This could be overcome by including negative sort after treatment by IAA in our protocol (see diagram *Figure 6—figure supplement 1I*). However, treatment with IAA resulted in 90–70% mCherry removal, which was close to the control level but not complete, limiting the effectiveness as complete removal tool (*Figure 6D*).

Rab11 family GTPases are implicated in cell migration, invasion, and cancer progression (*Kelly et al., 2012*; *Jin et al., 2021*; *Caswell et al., 2009*; *Caswell et al., 2007*; *Howe et al., 2020*). To test the functionality of DExCons/DExogrons, we performed a 2D scratch wound migration assay and found that Rab11a or Rab11b DExCon or DExogron modified cells closes the wound significantly later than control wt cells in the absence of dox-induced expression, and this phenotype is further amplified by Rab11a/b knock-out combinations (*Figure 6F*; *Figure 6—figure supplement 2A*; *Video 16*). Knock-out phenotypes were reversed when Rab11a/b expression was re-activated with dox; however, induction of degradation with IAA did not oppose this rescue, suggesting that IAA induced Rab11b depletion is incomplete in the presence of dox and is not sufficient to mimic the knock-out phenotype. The importance of Rab11a or Rab11b for cell migration was further verified with A2780 stably expressing shRNA anti-Rab11a or b (*Figure 6F*; *Figure 6—figure supplement 2A*,B). The effect of proliferation on wound healing closure was negligible, as any change in Rab11a/b expression did not alter cell growth (*Figure 6—figure supplement 2C*). Notably, we observed a toxic effect of IAA (100 µg/ml) from 10 hr of exposure when A2780 cells seeded sparsely.

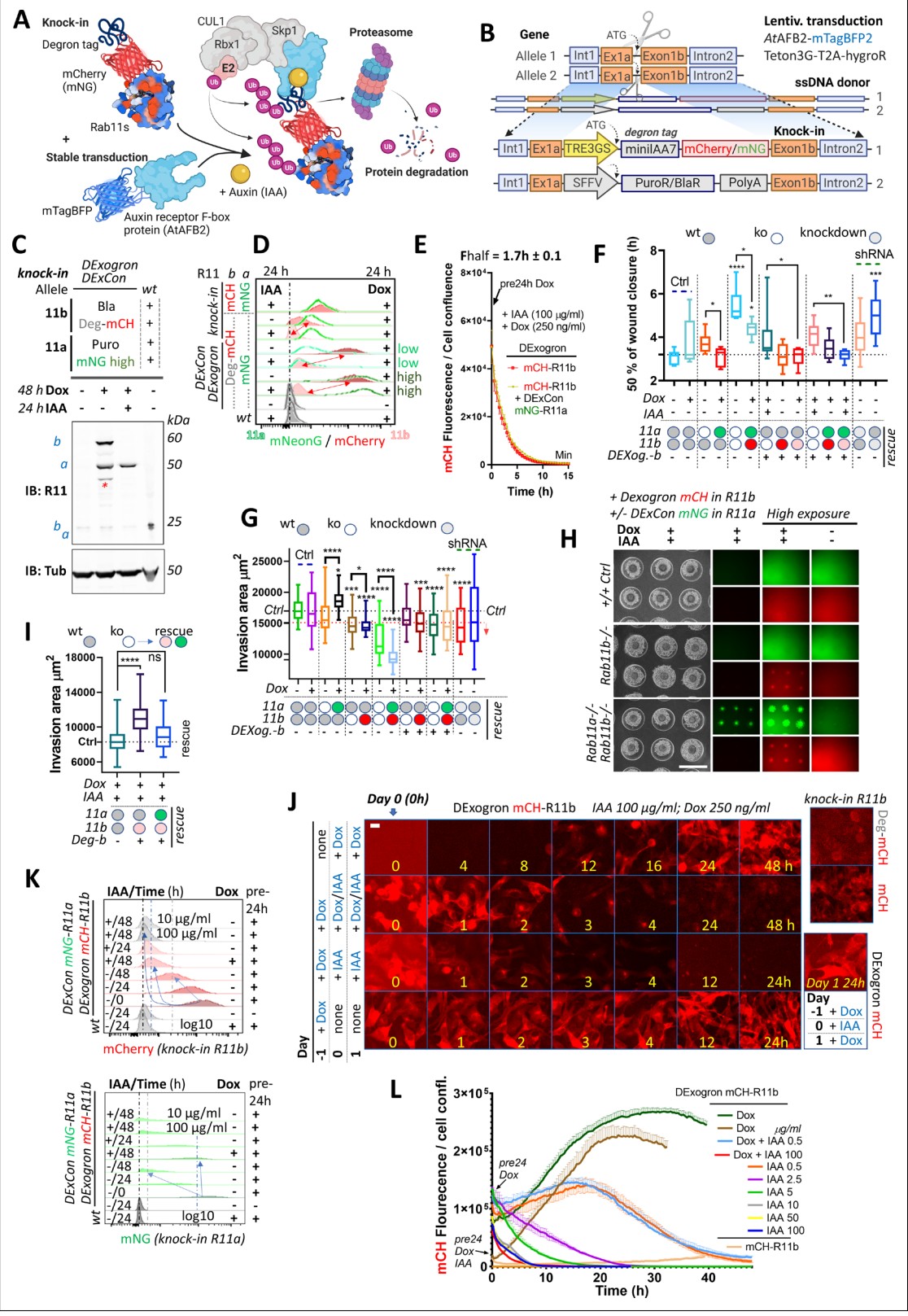

**Figure 6.** DExogron (DExCon combined with auxin-mediated targeted protein degradation): a tunable approach to modify protein levels within cells. (**A**) Schematic illustration of an improved auxin-inducible degron system with low-basal degradation (*Li et al., 2019*) used in this study. Degron tag=miniIAA7. (**B**) Schematic of components of knock-in cassette used for DExogron, together with second donor cassette (providing antibiotic resistance) independently targeting another allele. A2780 cells stably expressing AtAFBP2 and Teto3G-T2A-HygroR were used for knock-in;

*Figure 6 continued on next page*

*Figure 6 continued*

mCH=mCherry (red); mNG=mNeonGreen (green). All schematic illustrations were created with BioRender.com. (**C**) Immunoblots of mNeonGreen-Rab11a DExCon (sorted for high; the full blot with also cells sorted for low in *Figure 6—figure supplement 1F*)/miniIAA7(=Deg)-mCherry-Rab11b DExogron cells treated or not treated with doxycycline (dox) (250 ng/ml)±IAA (Indole-3-acetic acid as a source of auxin; 100 µg/ml). Bla/Puro represents Puromycin or Blasticidine resistance given by additional knock-in outcomes. Blots were probed with anti-Rab11 antibody targeting both Rab11a/b (Rab11). Tubulin (Tub), loading control. wt=unmodified A2780; a=Rab11a; b=Rab11b ±fluorophore knock-in. Stars indicate mCherry/iRFP670 lower molecular weight band caused by hydrolysis during sample preparation (*Gross et al., 2000*). (**D**) FACS analysis of DExogron-mCherry-Rab11b/DExCon-mNeonGreen-Rab11a (sorted for high or low) A2780 cells or classical Rab11a/Rab11b double knock-in cells±miniIAA7 (=Deg) without the TRE3GS promoter are shown. Cells were treated ±dox (250 ng/ml) ±IAA (100 µg/ml) for 24 hr as indicated. Red arrows indicate the dynamic range of mCherry intensity change. wt=unmodified A2780. Dashed line indicates negative Ctrl. Log10 scale shown. (**E**) Degradation kinetics of DExogron-mCherry-Rab11b (±DExCon-mNeonGreen-Rab11a; See also *Figure 6—figure supplement 1G*) analyzed by Incucyte imaging. Cells were pre-treated by dox for 24 hr followed by dox/IAA co-treatment as indicated and half-time of mCherry fluorescence decrease calculated. The mean of six replicates from three independent experiments is shown ± SEM. (**F**) Confluent cells were pre-treated for 24±dox (250 ng/ml)±IAA (100 µg/ml), scratch wounds introduced and imaged in phenol red-free RPMI ±dox/IAA as indicated. a-/- or b-/- knock-out shown as white circle; DExCon-mNeonGreen-Rab11a (rescue shown as green oval); DExCon- or DExogron(=Dexog.)-mCherry-Rab11b (rescue shown as red or pink oval indicating low mCherry expression) Box and whisker plot of 2D scratch wound migration (n=3–9 repeats from three independent experiments), shown as time required to close 50% of the normalized wound area. One-way ANOVA analysis Tukey post hoc test used for statistical analysis. (**G–I**) On-chip spheroid invasion assay of cells described in (F), migrating (72 hr) in collagen matrix supplemented with FN and treated ±dox (250 ng/ml)±IAA (100 µg/ml) as indicated (pre-induced for 24 hr). (**G**) Quantification of spheroid invasion ±dox (n=48–76 from three independent experiments). Black dotted line indicates Ctrl, red line is the threshold of statistical significance (see right arrow; statistics as described in F). See also *Figure 6—figure supplement 2D-F*. (**H**) Brightfield and fluorescence images of spheroids with lower or higher exposures (scale bar=1 mm) co-treated by ±dox/IAA and quantification of their invasion capacity (**I**). (**J**) Timelapse images of DExogron-mCherry-Rab11b A2780 cells (Incucyte, scale bar=20 µm). Cells were treated with dox/IAA as indicated. mCherry fluorescence intensity compared with classical endogenous Deg-mCherry-Rab11b or mCherry-Rab11b tagging (on the right). See *Video 14*. (**K**) FACS of DExCon/DExogron-Rab11a/Rab11b cells treated as indicated (dox 250 ng/ml; IAA 100 µg/ml if not stated otherwise). Pre-24 hr=dox pre-treatment for 24 hr prior analysis; wt=A2780 without CRISPR knock-in. Blue arrows and dashed lines indicate the rate at which signal intensity decreases. log10 scale is shown. (**L**) The effect of different dox, dox/IAA, or IAA levels (as indicated cells were dox or dox/IAA pre-treated for 24 hr) on fluorescence of DExogron-mCherry-Rab11b (conventional mCherry-Rab11b knock-in for comparison without any treatment; top right) analyzed by Incucyte S3 system from images as shown in (**J**). Representative curves of mCherry fluorescence normalized to cell confluence (three replicates ± SEM) are shown.

The online version of this article includes the following figure supplement(s) for figure 6:

**Figure supplement 1.** The DExogron (DExCon combined with auxin-mediated targeted protein degradation) module can tune protein levels within cells.

**Figure supplement 2.** DExCon (Doxycycline-mediated endogenous gene Expression Control)/DExogron (DExCon combined with auxin-mediated targeted protein degradation) reveal the contribution of Rab11a/b to migration and invasion.

**Figure supplement 3.** Characterization of DExCon-Rab11a/DExogron-Rab11b cells.

**Figure supplement 4.** Characterization of DExogron-Rab11a/DExCon-Rab11b cells.

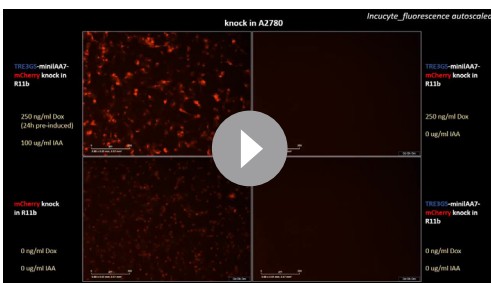

**Video 14.** Timelapse (Incucyte S3 system, 20×) of expression and degradation kinetics of miniIAA7-mCherry-Rab11b DExogron cells (A2780, comparison with the classical endogenous mCherry-Rab11b tagging) treated by ±doxycycline (dox ±Indole-3-acetic acid IAA) (from time 0 or cells pre-treated 24 hr with dox). Timelapse (mCherry channel is shown) covers total 49 hr with frame taken every 30 min (~30 min elapsed time per second of the movie); 96 well tissue culture plates (Corning); RPMI fenol-free media. Selected frames from this movie are shown in Figure 6J. https://elifesciences.org/articles/76651/figures#video14

To investigate how manipulation of Rab11a/b expression influences cell migration/invasion into 3D collagen/FN matrix, we performed spheroid invasion assays. Reducing Rab11a or b expression using shRNA, DExCon, DExogron, or their combination decreased the ability of cells/spheroids to invade FN-rich collagen, and this was significantly rescued by dox treatment in Rab11a DExCon single knock-in cells (*Figure 6G*; *Figure 6—figure supplement 2D,E*). Dox-induced re-expression of Rab11b had a dose-dependent dominant-negative effect on invasion under all combinations, sufficiently blocking even Rab11a rescue. The negative effect of Rab11b rescue was reversed by simultaneous dox/IAA co-treatment and led to cell invasion comparable to wt control cells or even significantly higher (*Figure 6H–I*) despite the cytotoxic effect of IAA observed on individually migrating cells (*Figure 6—figure supplement 2D,E* vs *Figure 6H*). This confirms

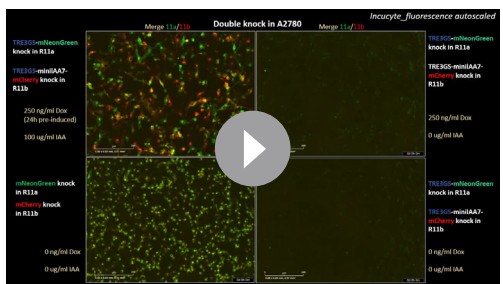

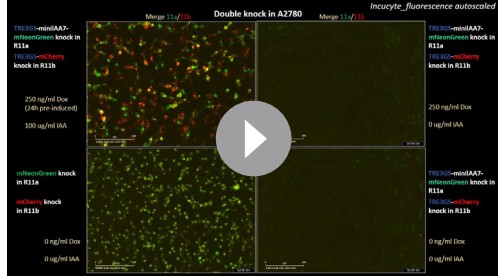

**Video 15.** Timelapse (Incucyte S3 system, 20×) of expression and degradation kinetics of miniIAA7-mCherry-Rab11b DExogron/mNeonGreen-Rab11a DExCon cells (A2780, comparison with the classical double endogenous mCherry/mNeonGreen-Rab11b/Rab11a tagging) treated by ±doxycycline (dox) ±Indole-3-acetic acid (IAA) (from time 0 or cells pre-treated 24 hr with dox). Timelapse (merge of mCherry/mNeonGreen channel is shown) covers total 49 hr with frame taken every 30 min (~30 min elapsed time per second of the movie); 96 well tissue culture plates (Corning); RPMI fenol-free media. Selected frames from this movie are shown in Figure 6—figure supplement 3A.

https://elifesciences.org/articles/76651/figures#video15

**Video 17.** Timelapse (Incucyte S3 system, 20x) of expression and degradation kinetics of miniIAA7-mNeonGreen-Rab11a DExogron/mCherry-Rab11b DExCon cells (clone of A2780, comparison with the classical double endogenous mCherry/mNeonGreen-Rab11b/Rab11a tagging) treated by ±doxycycline (dox) ±Indole-3-acetic acid (IAA) (from time 0 or cells pre-treated 24 hr with dox). Timelapse (merge of mCherry/mNeonGreen channel is shown) covers total 49 hr with frame taken every 30 min (~30 min elapsed time per second of the movie); 96 well tissue culture plates (Corning); RPMI fenol-free media. Selected frames from this movie are shown in Figure 6—figure supplement 4B.

https://elifesciences.org/articles/76651/figures#video17

Rab11b functionality and suggests that cells are sensitive to Rab11b re-expression levels and that balanced dosage is needed for coordination of membrane trafficking important for 3D cell migration.

We next tested whether induction of degradation with IAA can improve the AID pitfall of incomplete Rab11b depletion by withdrawing dox following re-activation of Rab11b expression. Indeed, mCherry fluorescence intensity of DExogron Rab11b cells or DExogron-Rab11b/DExCon-Rab11a cells was slightly, but significantly, decreased below the fluorescence of dox/IAA co-treated cells as early as 1–4 hr after switching to dox-free media supplemented only with IAA (*Figure 6J*; *Figure 6—figure supplement 3A*) and returned to non-detectable levels in 24–48 hr (*Figure 6K*). After 48 hr, expression of DExCon mNeonGreen-Rab11a was reduced, but still detectable (*Figure 6K*). These data show that the DExogron is an important modification that can lead to complete removal of protein following withdrawal of dox-induced gene expression.

Because IAA appeared to influence cell proliferation, we dropped its concentration from 100 µg/ml to 0.5 µg/ml and found that degradation was significantly slowed but still occurred to a similar extent up to 2.5 µg/ml (*Figure 6K and L*; *Figure 6—figure supplement 3B*). Dox/IAA co-treatment was sufficient to tune DExogron mCherry-Rab11b to endogenous levels (IAA 0.5 µg/ml; *Figure 6L*). DExogron-Rab11a/DExCon-Rab11b similarly resulted in tuned steady-state fluorescence intensity

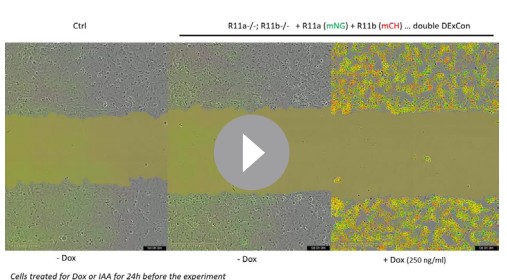

**Video 16.** A2780 (Ctrl; stably expressing AtAFBP2); Rab11s DExCons/DExogrons cells or cells with shRNA anti-Rab11a or b as indicated. Wound healing experiment automatically imaged and analyzed in real time by Incucyte S3 system (blue mask determined based on brightfield image taken at time 0 and every other frame taken after the scratch). Confluent cells were pre-treated for 24±doxycycline (dox) (250 ng/ml)±Indole-3-acetic acid (IAA) (100 µg/ml), scratch by wound the WoundMaker and imaged in RPMI fenol-free media ± dox/IAA as indicated. Timelapse (merge of brightfield with highlighted blue mask and true mNeonGreen/mCherry fluorescence) covers total 24 hr with frame taken every 1 hr (~1 hr elapsed time per second of the movie); ImageLock 96-well Plates; RPMI fenol-free media. Selected frames from this movie are shown in Figure 6—figure supplement 2A.

https://elifesciences.org/articles/76651/figures#video16

of mNeonGreen Rab11a with degradation halftime kinetics around 2.5±0.1 hr (*Figure 6—figure supplement 4A-E*; *Video 17*).

Because IAA induced depletion of DExogron Rab11b was incomplete when expression was driven in the presence dox, we pre-induced DExogron-Rab11b for 24 hr (*Figure 6—figure supplement 2F*) and then transferred cells to dox-free media with IAA (10 g/ml) for 24 hr before scratch. As expected, this condition led to complete mCherry removal (not detectable by fluorescent microscopy) and was sufficient to reverse the 'rescued' motility of DExogron-mCherry-Rab11b or DExogron-mCherry-Rab11b/DExCon-mNeonGreen-Rab11a cells (*Figure 6—figure supplement 2F*).

These data therefore indicate that DExogron can eliminate the shortcomings of the AID system, allowing fine-tuning of expression levels up to complete depletion following CRISPR-based modification. Furthermore, our results suggest that although Rab11b and Rab11a may have a complementary function in migration and invasion of A2780 ovarian cancer cells, they differ substantially in their post-transcriptional regulation.

## Tuning Rab11 expression levels with DExCon and DExogron modulates TFN receptor recycling

Recycling of TFN-R is typically biphasic with an initial rapid direct route from early endosomes to the plasma membrane (under control of Rab4) and a slower non-essential route that requires additional trafficking through perinuclear recycling endosomes via Rab11 (*Daro et al., 1996*; *Sheff et al., 2002*; *Sheff et al., 1999*; *Figure 7A*). Our previous experiments demonstrated that the DExCon/DExogron system induces expression of Rab11 family members to promote cell motility (*Figure 3N*; *Figure 6F–H*), the expressed proteins show endogenous localization within cells (2E; *Figure 5—figure supplement 2A-D*) and they traffic together with TFN (*Figure 2—figure supplement 2D*; *Video 3*), suggesting that they are fully functional. We further tested how DExCon/DExogron controlled Rab11 family expression level influenced recycling of internalized TFN to test the function of these re-expressed proteins in a different context. DExCon or DExogron modified Rab11a or Rab11b cells which lack expression of the targeted GTPase(s) had little impact on TFN-R recycling at 30 min (*Figure 7B*), suggesting that fast recycling is likely predominant in A2780 cells as in other cell types (*Sheff et al., 1999*). Re-expression of Rab25 did not impact upon recycling, indicating that this Rab11 family member may have a limited influence on trafficking this cargo (*Figure 7B*). However, dox-induced high-level expression of either Rab11a or Rab11b under DExCon/DExogron control decreased the level of recycling at 30 min (*Figure 7B*), consistent with the literature (*Ren et al., 1998*), perhaps by favoring the slow-recycling route. DExCon/DExogron Rab11a/b cells selected for expression levels comparable to endogenous Rab11a/b knock-in did not have any significant effect on TFN-R recycling rate compared to the non-modified control, indicating that it is the expression level that slows recycling. Similarly, IAA/Dox co-treatment of DExogron-Rab11b cells led to near-physiological levels of Rab11b which had no significant effect on TFN-R recycling (*Figure 7B*). Surprisingly, TFN-R recycling slowed by high Rab11a DExCon levels was restored to the level of double knock-out cells by lowering Rab11b levels with DExogron (Rab11a DExCon/Rab11b DExogron, IAA/dox co-treatment). These data suggest that Rab11a/Rab11b can recycle TFN-R via the slow recycling pathway, and that increasing levels of Rab11a/b could favor the slow-recycling route to delay recycling of TFN.

To test the effect of an acute decrease of Rab11a/Rab11b levels toward the expression driven by endogenous locus, we pre-adapted A2780 cells to high Rab11a/Rab11b levels by treating them with dox over >7 days before decreasing Rab11 levels by switching to IAA; dox+IAA or dox-positive media for 24–48 hr prior analysis. TFN-R recycling was assessed after 10 min to highlight competition between fast and slow recycling routes (*Figure 7C*). Cells induced to have high-level expression of DExCon-mNeonGreen-Rab11a or/and DExogron-mCherry-Rab11b again showed lower levels of TFN-R recycling compared to control (mainly evident for Rab11a), but tempering expression of Rab11a and Rab11b (or Rab11b alone in DExogron-mCherry-Rab11b cells) by withdrawing dox/adding IAA demonstrated that lower Rab11 levels correlated with increased recycling rates (7C). These data further support the notion that competition between fast and slow recycling likely controlled by Rab4 and Rab11a/b, respectively, underpins the differences in recycling rates induced by increasing Rab11 expression level.

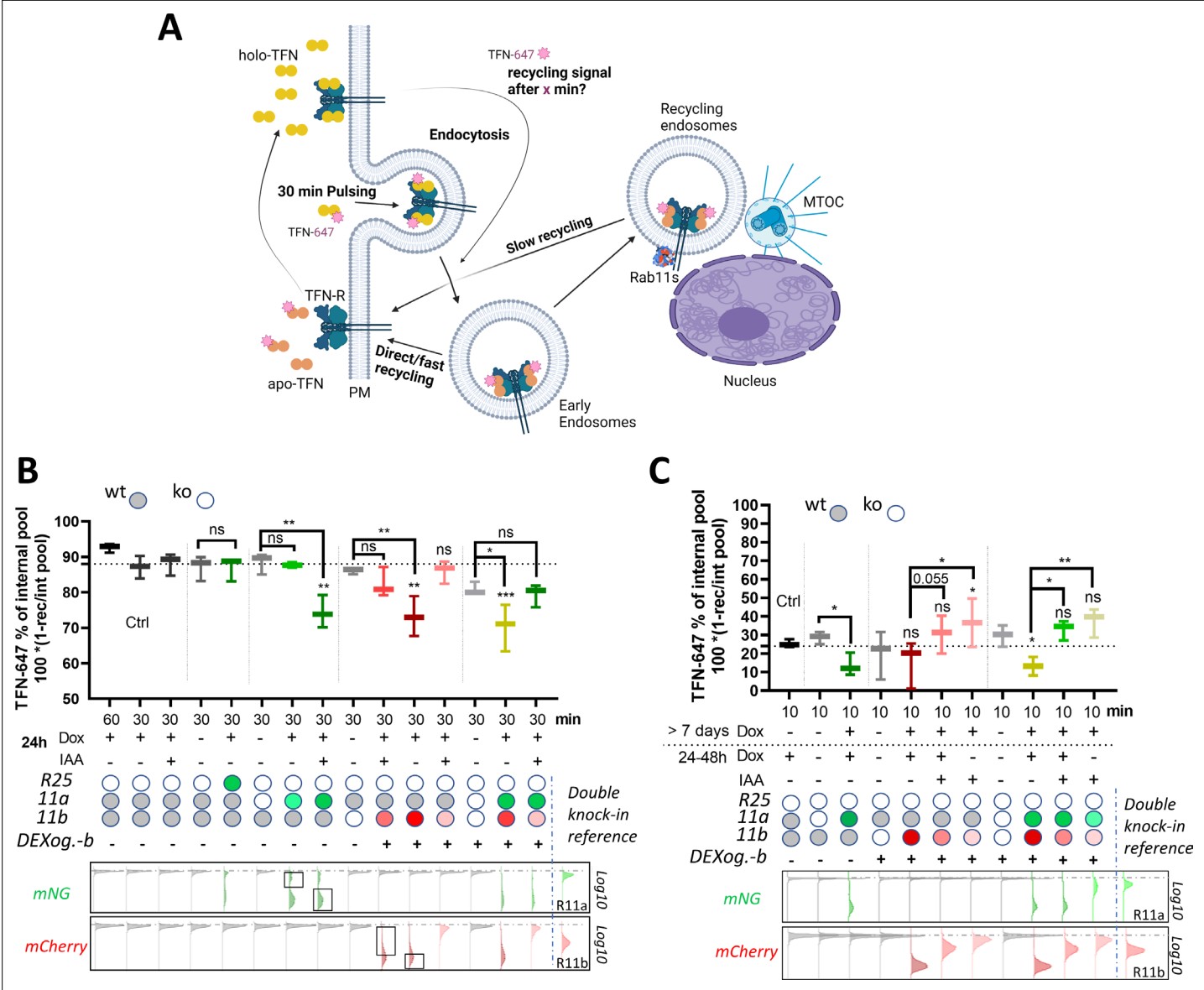

**Figure 7.** Rab11 expression levels modulate transferrin (TFN) receptor recycling. (**A**) Schematic illustration (created with Biorender.com) of TFN-647 recycling assay with depicted pathways: rapid direct route from early endosomes to the plasma membrane and a slower/longer route through perinuclear recycling endosomes via Rab11. (**B–C**) TFN-647 recycling assay with A2780 DExogron-mCherry-Rab11b, DExCon-mNeonGreen-Rab11a cells or their combination described previously in *Figure 6*, mNeonGreen-Rab25 DExcon cells sorted for high as shown in *Figure 3—figure supplement 1D*. Cells doxycycline (dox) pre-treated follow by ±dox/Indole-3-acetic acid (IAA) or their combinations as described; dox (250 ng/ml); IAA (100 in **B** or 10 µg/ml in **C**). TFN-647 recycled levels were measured at 30 min (**B**) or 10 min (**C**) by flow cytometry and normalized to TFN-647 internalization levels (reached in 30 min) as % of internal pool. IAA or dox/IAA condition in (**C**) required extra normalization based on the wt control to subtract the negative effect of IAA. wt=CRISPR unmodified A2780 cells stably expressing AtAFBP2 and Teto3G-T2A-HygroR. FACS analysis of mNeonGreen/mCherry fluorescence levels shown below graphs (**B–C**), dashed line indicates negative Ctrl. Log10 scale shown. Black rectangles indicate subpopulations (high/low) analyzed for TFN-647 recycling rate. Classical Rab11a/Rab11b double knock-in cells used as reference for mNeonGreen/mCherry brightness. Box and whiskers (geometrical mean) are shown from three independent biological replicates (**B–C**). One-way ordinary or repeated-measures ANOVA analysis Tukey post hoc test used for statistical analysis and compared to Ctrl (**B**; 30 min recycling) or among different conditions across modified Rab11a/b/25 toward untreated cells or as indicated (**B–C**).

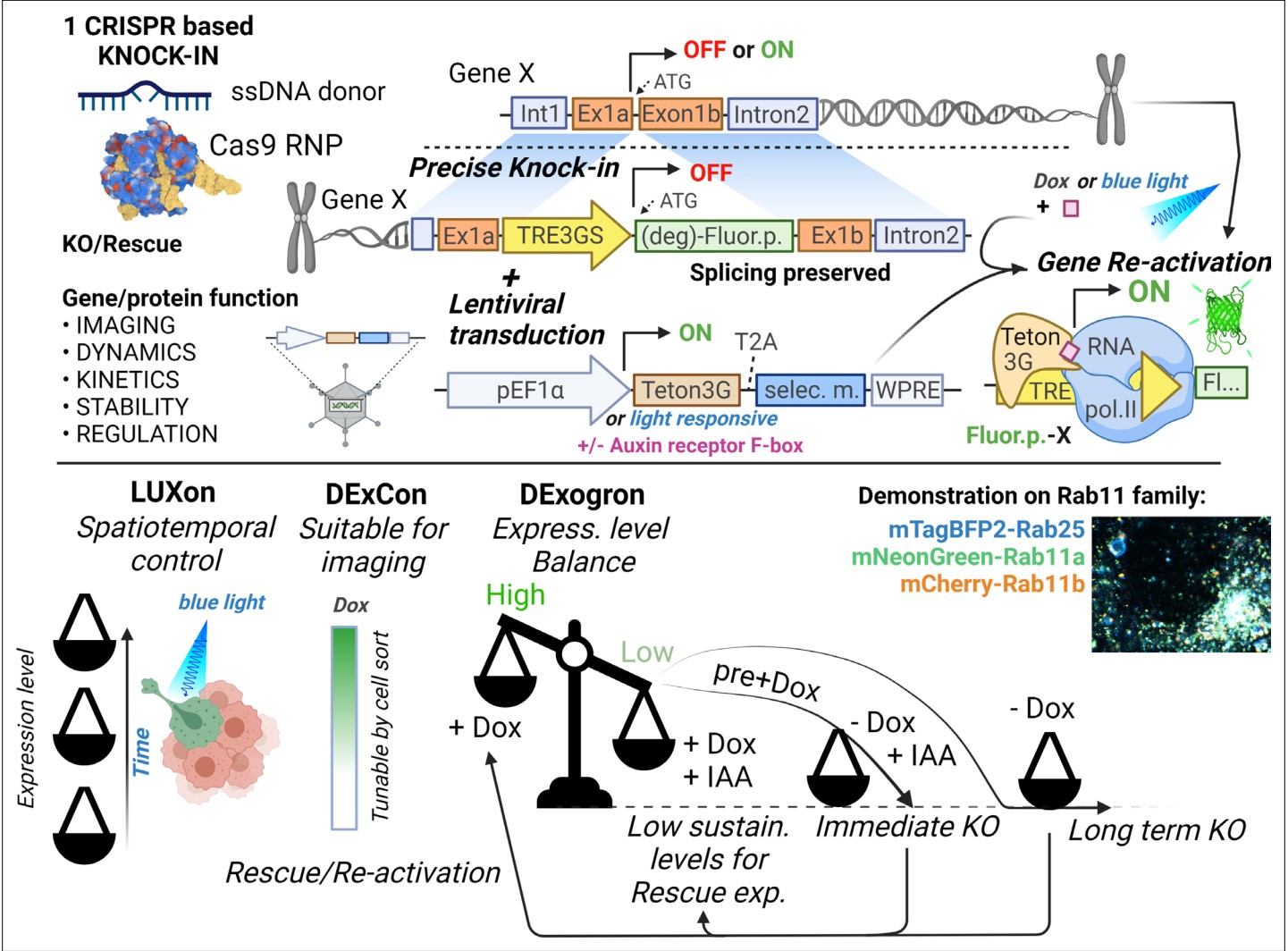

**Figure 8.** Tools for on-demand expression control of endogenous genes. Schematic illustration (created with Biorender.com) of DExCon(Doxycycline-mediated endogenous gene Expression Control), DExogron (DExCon combined with auxin-mediated targeted protein degradation), and LUXon (light responsive DExCon) approaches, where a single CRISPR/Cas9-mediated gene editing event can block endogenous gene expression, with the ability to re-activate expression encoded such that even silent genes can be expressed. Expression can be controlled systematically using doxycycline, or spatiotemporally by light, tuned to physiological levels by auxin, allowing fluorescent tagging of endogenous proteins and quantification of expression kinetics, protein dynamics, and stability for highly similar genes such as members of the Rab11 family.

Taken together, this suggests that the DExCon/DExogron system affords control over the timing and level of gene expression of Rab11 family members, and points toward a more complex relationship between fast Rab11-independent and slow Rab11-dependent recycling.

## Discussion

Here, we describe DExCon, LUXon, and DExogron tools (see *Figure 8*) to reversibly modulate gene expression from the endogenous locus. These systems allow effective knock-out of gene expression, but allow subsequent re-activation of transcription (or activation of transcriptionally silent genes). LUXon permits the spatial and temporal control of gene expression, and DExogron adds a further level of control such that expression levels can be tuned appropriately, or complete degradation induced.

Significant effort has been made to achieve conditional control of gene expression at the endogenous locus in order interrogate gene function in a native biological and pathological context. Binary recombination approaches, such as the Cre/loxP system, allow inducible inactivation (*Quadros et al.,*

*2017*) or activation through conditional excision of a stop element in the target gene (*Ventura et al., 2007*). However, the ability to control gene expression is lost following irreversible deletion, and this system does not allow evaluation of the phenotypic effects across a spectrum of different expression levels. Our DExCon/Dexogron/LUXon approaches are attractive alternatives as they offer, in addition to reversible inactivation of protein-coding genes, conditional and (for LUXon) spatiotemporal expression control of re-activated genes which are visualized by an introduced FP.

Other approaches to modulate endogenous gene expression have used components of lac repression and tet activation systems introduced to the endogenous promoter sequence or downstream, including 'Reversible Manipulation of Transcription at Endogenous loci-control' (*Lee et al., 2017*). Our approach differs because it bypasses the function of the endogenous promoter and does not require insertion of effector binding sites that can perturb target gene expression. Our one-step CRISPR knock-in based method is best suited for any genes whose protein-coding transcripts share the initial ATG codon and can be N-terminally tagged, or introduce the DExCon module upstream of alternatively edited genes. For genes where the fluorophore would have to be introduced internally or C-terminally, two-step CRISPR knock-in approaches are necessary (one for TRE3GS promoter, second for FP). Although our system does not silence transcription driven by an endogenous active promoter, it generates non-sense frameshifts in transcripts originating from the endogenous promoter and can activate silenced transcripts. This offers advantages over methods that alter promoter methylation (e.g. dCas9 fused to transcriptional activators or repressors *Pflueger et al., 2019*, which can create modifications over several tens of kilobases and impact off-target genes *Peng et al., 2009*; *Groner et al., 2010*) and overcomes the limit of classical knock-in where low expression confounds detection or protein products. Furthermore, in contrast to transient transfection of exogenous vectors, DExCon preserves splicing control and allows analysis of the full complement of physiologically localized isoforms within cells (*Figure 2E–G*; *Figure 2—figure supplement 2A,B*; *Figure 5—figure supplement 2A-D*).

The AID system is an excellent method of protein down-regulation (*Nishimura et al., 2009*), which has since been further developed (*Röth et al., 2019*; *Li et al., 2019*; *Yesbolatova et al., 2020*) allowing rapid degradation of target proteins. Using CRISPR, we successfully applied the improved AID system to the endogenous locus of Rab11b (*Li et al., 2019*); however, the incorporation of degron miniIAA7 tag in front of mCherry at the Rab11b locus resulted in a profound decrease in mCherry fluorescence without induction by IAA, and induction did not lead to complete protein removal (*Figure 6D*). Our combination of degron and DExCon (DExogron), however, can generate robust expression and allows additional fine temporal control of the DExCon system, providing a system that can reveal acute phenotypes that otherwise might be masked by compensation or a dominant terminal phenotype.

For broad application of strategies such as DExCon, feasibility and efficiency are critical. One of the main obstacles to gene editing via HDR is its low efficiency, which complicates targeting all alleles, generation, and selection of homozygous knock-in modifications (*Gurumurthy et al., 2019*). Our optimized magnetofection mediated delivery of Hifi Cas9 as a RNP complex with single-stranded donor DNA ensured precise and on-target integrations to endogenous Rab11 family gene loci reflected by expected localization in cells (*Figure 1C*). Although truncated integration events are often reported (*Canaj et al., 2019*), functional flanking components (tet-responsive promoter and FP) allow selection of polyclonal populations with full integration of the ssDNA template (including for example the nanobody anti-GFP domain), negating the need for comprehensive and labor-intensive genotyping (*Figure 2H–I*). We demonstrated the robustness of our approach by generating single, double, and triple knock-in DExCon at the endogenous loci of Rab11 family genes in A2780 cells without the need for clonal selection. Our strategy further offers sufficient expression modulation from monoallelic modifications; therefore, we could independently target alleles of the same gene using combinations of different donors (DExCon-mNeonGreen, DExCon-mCherry, antibiotic resistance [*Figure 5F*; *Figure 6—figure supplement 1B-F*]). As proof of principle, we imaged protein products of both Rab11b alleles to demonstrate new possibilities, for example, studying the dynamics of mutant and wildtype alleles of oncogenes and tumor suppressors. When simultaneously delivering DExCon together with antibiotic resistance to a second allele, we observed a dramatic increase in selection efficiency from 0.2 to 25% (FP-positive dox-responsive cells surviving antibiotic treatment) allowing for easy enrichment of homozygous edits (*Figure 6—figure supplement 1E*). Sequential delivery of antibiotic resistance to DExCon-Rab11a and -Rab11b (*Figure 6—figure supplement 1B-F*)

further improved editing outcome, effectively generating a knockout of one allele to ensure that all protein generated by cells from the second allele is tagged appropriately (mNeonGreen/mCherry in this case), full knock-out with dox-controlled expression. The conventional Tet-OFF/ON system is widely used in various research models, such as transgenic animals (*Zhu et al., 2002*) and can be easily combined with our DExCon/DExogron/LUXon strategy for a huge range of applications across the field of biology. Notably, DExogron harbors promising application also for essential genes with knock-out lethal phenotypes. Here, the continual presence of dox or the transition from Tet-on to Tet-off system would be required, and IAA treatment could induce immediate knock-out effect.

To demonstrate the potency of our strategies on closely related genes, we deployed them to study members of the Rab11 family (Rab11a, Rab11b, and Rab25) in an ovarian cancer cell line that lacks endogenous Rab25 expression (*Figure 3A*). We demonstrated that spatiotemporal re-activation of silenced endogenous Rab25 expression can promote invasive migration (*Figure 3J–O*), and that Rab11b and Rab11a may have complementary functions in migration and invasion (*Figure 6F–I*). We identified and reversibly rescued two Rab11b protein-coding splice variants (201 [24.5 kDa] and 202 [20 kDa]) in a physiologically proportional manner (*Figure 2—figure supplement 1B*). Mutants of Rab11 that lack the C-terminus, similar to Rab11b variant 202, have previously been shown localize diffusely and are unable to bind and recycle TFN-R (*Schlierf et al., 2000*). However, we observed specific endosomal localization of CRISPR/Cas9-modified Rab11b (*Figure 2E*; *Figure 5—figure supplement 2B*) likely corresponding with the 201 variant and suggesting that the 202 variant is not a significant protein product. Interestingly, cells migratory/invasive responses appear to be sensitive to expression levels, particularly for Rab11b. Cells handle recycling cargoes by numerous routes simultaneously, and these differences could be explained by favoring trafficking of a cargo along Rab11b routes by increasing expression, for example, Rab11/Arf6 and Rab4-dependent trafficking of $\alpha5\beta1$ and $\alpha v\beta3$ integrins appear to act as competing pathways in migrating cells to modulate migration (*Caswell et al., 2008*; *Morgan et al., 2013*). The tunable Rab11 family expression control further revealed a pleiotropic effect of Rab11a/b, and Rab25, on their recognized cargo TFN (*Figure 7A–C*) and further suggests that cells may be sensitive to Rab11 re-expression levels that can change the coordination of membrane trafficking (*Daro et al., 1996*; *Sheff et al., 2002*; *Sheff et al., 1999*). Rab11a/b seem to have little influence on the rate of TFN recycling when knocked-out by DexCon/DExogron, suggesting that Rab4 rapid recycling (or other routes) predominate in the cells used in this study (*Figure 7B and C*). The ability of Rab11a/b to divert TFN via a longer recycling route to delay return to the surface likely stems from TFN receptor carrying capacity, as higher DexCon/DExogron Rab11 levels have physiological endosomal membrane localization (*Figure 2E*; *Figure 5—figure supplement 2A-D*). Interestingly, lowering/tuning Rab11b expression restored TFN-R recycling rate slowed by high Rab11a levels (*Figure 7B and C*), and Rab25 expression, even at high level, has no discernible effect on TFN recycling (*Figure 7A*), as found previously in MDCK cells *Casanova et al., 1999*. Rab11a/b/25 colocalize with TFN to a similar extent (*Figure 2—figure supplement 2D, E*), and it is therefore interesting to speculate that this could be underpinned by their different relocalization/recycling kinetics (*Figure 5M*) or by differences in effector recruitment pathways/cargo selectivity.

DExCon/DExogron tools allowed us to quantify surprising differences between Rab11 family members. We found that our system reflects the accessibility for transcriptional machinery to drive transcription at the endogenous locus, demonstrating that Rab11a has significantly faster expression kinetics and compared to Rab11b and re-activated Rab25 expression is slower still (*Figure 5A–C*). However, in cells with endogenous Rab25 expression, expression kinetics are demonstrable faster (*Figure 5D*). We also demonstrated that the ability to relocalize differed significantly between Rab11 family members with Rab25 as the least dynamic (*Figure 5K–M*; *Figure 5—figure supplement 3A-C*). The faster relocalization rate of Rab11a over Rab11b could be explained by its more diffuse localization (*Figure 5—figure supplement 2C,D*), which could suggest it is less stably associated with endosomal membranes. Furthermore, we observed a population of Rab25 specific vesicles in the nuclear periphery which appeared to be less mobile (*Figure 3I*), and these vesicles were always negative for TFN receptor (*Figure 4B–F*). Such vesicles with limited mobility were recently described to be located on the nuclear envelope, where they exhibit propensity to undergo fusion and act as a novel endosomal route from the cell surface to the nucleoplasm that facilitates the accumulation of extracellular and cell surface proteins (e.g. epidermal growth factor receptor (EGFR)) in the nucleus (*Shah et al., 2019*; *Chaumet et al., 2015*). Rab25-mediated EGFR recycling (*Caswell et al., 2007*; *Caswell et al.,*

*2008*) and elevated levels of nuclear EGFR correlate with poor prognosis in ovarian cancer (*Xia et al., 2009*), and our data which suggest that Rab25 may be connected with nuclear envelope-associated endosomes warrants further investigation.

Overall, the work presented here opens new avenues not only for the precise expression control of endogenous genes, but also for the manipulation and imaging of gene function. All the benefits of gene editing can be obtained in one CRISPR based knock-in. DExCon, DExogron, and LUXon are powerful tools that allow new insight into the function of individual genes and closely related gene families.

# Materials and methods

## Key resources table

| Reagent type (species) or resource | Designation | Source or reference | Identifiers | Additional information |
|---|---|---|---|---|
| Cell line (*Homo sapiens*) | Ovarian cancer | ATCC | OVCAR-3 | Cell line maintained in RPMI-1640 media |
| Cell line (*H. sapiens*) | Ovarian cancer | DOI:10.1038/nm1125 | A2780-DNA3 | Cell line maintained in RPMI-1640 media |
| Cell line (*H. sapiens*) | Ovarian cancer | ECACC | COV362 | Cell line maintained in Dulbecco's Modified Eagles Medium (DMEM) (D5796) media |
| Cell line (*H. sapiens*) | Telomerase immortalized fibroblasts | DOI:10.1016/j.devcel.2007.08.012 | Telomerase immortalized fibroblasts | Cell line maintained in DMEM (D5796) media |
| Cell line (*H. sapiens*) | Embryonic kidney cells | ATCC | human embryonic kidney (HEK)293T | Cell line maintained in DMEM (D5796) media |
| Antibody | anti-Rab11 (Rabbit polyclonal) | Thermofisher | Cat# PA5-31348 | WB (1:1000) |
| Antibody | anti-α-tubulin (Mouse monoclonal) | Abcam | Cat# ab7291 | WB (1:25000) |
| Antibody | anti-Rab11a (Rabbit polyclonal) | Cell signaling (CST) | Cat# 2,413 S | WB (1:1000) |
| Antibody | anti-Rab25 (Rabbit monoclonal) | Cell signaling (CST) | Cat# 13,048 S | WB (1:1000) |
| Antibody | anti-RFP (5F8) (Rat monoclonal) | Chromotek | Cat# 5f8-100 | IF(1:200), WB (1:750) |
| Antibody | anti-mNeonGreen (Mouse monoclonal) | Chromotek | Cat# 32F6-100 | IF(1:200), WB (1:400) |
| Transfected construct (human) | pCDH-antiGFPnanobody-mCherry-Rab11a | this paper | | Lentiviral construct |
| Transfected construct (synthetic) | PAtetOFF_L317_C1634=_ CSII-CAG-MCS-IRES-mCherry | DOI:10.1016/j.celrep.2018.09.026 | | Lentiviral construct for photoactivatable TetOFF system |
| Transfected construct (synthetic) | PAtet2OFF_L395_CSII-CAG=T89 T2A=MCS-IRES-Snap-Bsd | DOI:10.1016/j.celrep.2018.09.026 | | Lentiviral construct for photoactivatable Tet2OFF system |
| Transfected construct (synthetic) | PAtetON_L321_C1673=_ CSII-CAG-MCS-IRES-Snap-Bsd | DOI:10.1016/j.celrep.2018.09.026 | | Lentiviral construct for photoactivatable TetON system |
| Transfected construct (synthetic) | pLenti Lifeact-iRFP670 BlastR | DOI:10.1038/s41467-018-05367-2 | RRID:Addgene_84385 | Lentiviral construct to transfect and express LifeAct. |

*Continued on next page*

*Continued*

| Reagent type (species) or resource | Designation | Source or reference | Identifiers | Additional information |
|---|---|---|---|---|
| Transfected construct (synthetic) | pLenti Lifeact-mTagBFP2 PuroR | Ghassan Mouneimne lab | RRID:Addgene_101893 | Lentiviral construct to transfect and express LifeAct. |
| Transfected construct (human) | pGIPZ-PuroR-IRES-GFP_shRNA anti Rab11b | Dharmacon | Cat# RHS4531-EG9230 clone id:V3LHS_365285 | Lentiviral construct to transfect and express the shRNA. |
| Transfected construct (human) | pGIPZ-PuroR-IRES-GFP_shRNA anti Rab11a | Dharmacon | Cat# RHS4430-200301210 clone id:V3LHS_411101 | Lentiviral construct to transfect and express the shRNA. |
| Recombinant DNA reagent | pCDH-TRE3GS-EF1a-tagBFP-T2A-TetOn3G | Andrew Gilmore lab | | Lentiviral tetracycline inducible backbone |
| Transfected construct (synthetic) | pCDH-EF1a-tagBFP-T2A-TetOn3G | this paper | RRID:Addgene_179888 | Lentiviral construct to express TetOn3G |
| Transfected construct (synthetic) | pCDH-EF1a-HygroR-T2A-TetOn3G | this paper | RRID:Addgene_179887 | Lentiviral construct to express TetOn3G |
| Transfected construct (human) | mCherry-Rab11a | Michael Davidson lab | RRID:Addgene_55124 | Mammalian Expression vector for transfection |
| Transfected construct (human) | GFP-Rab11b | Marci Scidmore lab | | Mammalian Expression vector for transfection |
| Transfected construct (human) | GFP-Rab11a | DOI:10.1016/j.devcel.2007.08.012 | | Mammalian Expression vector for transfection |
| Transfected construct (*Arabidopsis thaliana*) | pSH-EFIRES-P-AtAFB2 | DOI:10.1038/s41592-019-0512-x | RRID:Addgene_129715 | Mammalian Expression, CRISPR, and TALEN |
| Transfected construct (*A. thaliana*) | pCDH-AtAFB2-mTagBFP2 | this paper | RRID:Addgene_179889 | Lentiviral construct |
| Transfected construct (synthetic) | pMito-mCherry-FRB | DOI:10.1242/jcs.124834 | RRID:Addgene_59352 | Mammalian Expression vector for transfection |
| Transfected construct (synthetic) | GFP-FKBP | Stephen Royle lab | | Mammalian Expression vector for transfection |
| Transfected construct (synthetic) | pMito-iRFP670-FRB | this paper | | Mammalian Expression vector for transfection |
| Transfected construct (human) | pJET-50-DExogron-mCherry-R11b | this paper | RRID:Addgene_179904 | CRISPR and knock-in donor |
| Transfected construct (human) | pJET-49-DExogron-mNeonGreen-R11a | this paper | RRID:Addgene_179903 | CRISPR and knock-in donor |
| Transfected construct (human) | pJET-42-DExCon-antiGFPnanobody-mCherry-R11a | this paper | RRID:Addgene_179902 | CRISPR and knock-in donor |

*Continued on next page*

*Continued*

| Reagent type (species) or resource | Designation | Source or reference | Identifiers | Additional information |
|---|---|---|---|---|
| Transfected construct (human) | pJET-41-DExCon-antiGFPnanobody-mCherry-R11b | this paper | RRID:Addgene_179901 | CRISPR and knock-in donor |
| Transfected construct (human) | pJET-59-DExCon-mTagBFP2-R25 | this paper | RRID:Addgene_179900 | CRISPR and knock-in donor |
| Transfected construct (human) | pJET-67-DExCon-mNeonGreen-R11b | this paper | RRID:Addgene_179899 | CRISPR and knock-in donor |
| Transfected construct (human) | pJET-23-DExCon-mCherry-R11b | this paper | RRID:Addgene_179898 | CRISPR and knock-in donor |
| Transfected construct (human) | pJET-22-DExCon-mNeonGreen-R11a | this paper | RRID:Addgene_179897 | CRISPR and knock-in donor |
| Transfected construct (human) | pMKRQ_11_DExCon-mNeonGreen-R25 | this paper | RRID:Addgene_179896 | CRISPR and knock-in donor |
| Transfected construct (human) | pJET-64-SFFV-T2A-BlaR-polyA-R11b | this paper | RRID:Addgene_179895 | CRISPR and knock-in donor |
| Transfected construct (human) | pJET-63-SFFV-T2A-BlaR-polyA-R11a | this paper | RRID:Addgene_179894 | CRISPR and knock-in donor |
| Transfected construct (human) | pJET-62-SFFV-T2A-PuroR-polyA-R11b | this paper | RRID:Addgene_179893 | CRISPR and knock-in donor |
| Transfected construct (human) | pJET-61-SFFV-T2A-PuroR-polyA-R11a | this paper | RRID:Addgene_179892 | CRISPR and knock-in donor |
| Transfected construct (human) | pJET-10-mCherry-R11b | this paper | RRID:Addgene_179891 | CRISPR and knock-in donor |
| Transfected construct (human) | pJET-07-mNeonGreen-R11a | this paper | RRID:Addgene_179890 | CRISPR and knock-in donor |
| Transfected construct (human) | pJET-60-mNeonGreen-R11b | this paper | | CRISPR and knock-in donor |
| Transfected construct (human) | pJET-24-DExCon-antiGFPnanobody-mCherry-R25 | this paper | | CRISPR and knock-in donor |
| Transfected construct (human) | pJET_28_mCherry-minilaa7-Rab11b | this paper | | CRISPR and knock-in donor |
| Chemical compound and drug | doxycycline hydrochloride | Sigma | #D3447 | |
| Chemical compound and drug | SiR-Tubulin | Cytoskeleton | Cat# CY-SC002 | |

*Continued on next page*

*Continued*

| Reagent type (species) or resource | Designation | Source or reference | Identifiers | Additional information |
|---|---|---|---|---|
| Peptide, recombinant protein | Alt-R S.p. HiFi Cas9 Nuclease V3 | IDT | Cat#: 1081061 | |
| Chemical compound and drug | Lipofectamine CRISPRMAX Cas9 Transfection Reagent | Thermo | CMAX00008 | |
| Peptide, recombinant protein | Transferrin labeled 647 1 mg | Thermo | #T23366 | |
| Chemical compound and drug | Indole-3-acetic acid sodium salt | Cambridge Biosc. | 16954–1 g-CAY | |
| Chemical compound and drug | Rapamycin | Sigma | #R8781-200UL | |
| Sequence-based reagent | tracrRNA | IDT | Cat#: 1072532 | |
| Sequence-based reagent | crRNA_Rab11a | IDT | crRNA | GGTAGTCGTACTCGTCGTCG |
| Sequence-based reagent | crRNA_Rab11b | IDT | crRNA | CCGGAAGCGCCAGGACAATG |
| Sequence-based reagent | crRNA_Rab25 | IDT | crRNA | CCTCCATGCGGAGCCAAGAT |
| Software and algorithm | Macro for relocalization to mitochondria | this paper | | https://doi.org/10.6084/m9.figshare.17085632 |
| Software and algorithm | Macro for spheroid invasion assay | this paper | | https://doi.org/10.48420/16878829 |
| Software and algorithm | Python script for colocalization mapping | this paper | | https://doi.org/10.6084/m9.figshare.16810546 |

Detailed list of antibodies, chemicals, sequences, plasmids, and resources used are described in supplementary excel file S1 or accessible from https://doi.org/10.48420/14999550.

## Constructs

A range of plasmid DNA constructs were used and generated in this study including 27 donors used for long ssDNA preparation (available from Addgene: number 179887–179904), and these are listed with details in excel file S1. Plasmid maps are provided here https://doi.org/ 10.48420/14999526 (donors) and https://doi.org/10.48420/16810525 (others). Online tool Benchling was used to design all donors (https://www.benchling.com) and guideRNA sequences for specific CRISPR/Cas9 cleavage.

To prepare plasmid donors for ssDNA synthesis, first cDNAs coding for mNeonGreen/mCherry/ iRFP670/antiGFPnanobody/minilAA7/TRE3GS/SFFV-T2A-PuroR-polyA (and so on) or gene fragments (Rab11a/b/25) designed with universal linkers (including SpeI/AgeI or MluI site; see plasmid maps and catalog of donors in excel file S1) were commercially synthesized by GENEWIZ (FragmentGENE), ThermoFisher (geneArt Gene Synthesis), or IDT (gBlocks) depending on the synthesis limitations (GC-rich/repetitive sequences; see excel file S1). These DNA sequences were delivered in pUC57 or pMKRQ vectors or cloned into pJET1.2 vector (CloneJET PCR Cloning Kit. #K1231; Thermofisher) and later combined using SpeI/AgeI/XhoI sites or by Gibson assembly. Gibson assembly and SpeI/AgeI/MluI/EcoRI sites were also used to add, switch, or remove additional sequences (e.g. mTagBFP2 from pLenti Lifeact-mTagBFP2_PuroR). Lentiviral pCDH-antiGFPnanobody-mCherry-Rab11a (overexpressing control) was cloned using pCDH-EF1-tagBFP-T2A-mycBirA*-Rab11a (previously prepared in our lab) via SpeI/SalI and XbaI/XhoI complementary cleavage sites. pMito-mCherry-FRB (*Cheeseman et al., 2013*), a gift from Stephen Royle (Addgene plasmid #59352), was used as a template for generation of pMito-iRFP670-FRB (used with GFP-FKBP for knocksideways). pSH-EFIRES-P-AtAFB2

(Addgene plasmid #129715) (*Li et al., 2019*), pCDH-EF1-tagBFP-T2A-mycBirA*-Rab11a, and mTagBFP2 were used to generate lentiviral pCDH-AtAFB2-mTagBFP2. TRE3GS was removed by Cla/EcorI cleavage from third-generation tet inducible vector pCDH-TRE3GS-EF1a-tagBFP-T2A-TetOn3G (gift from Andrew Gilmore lab) to generate DExCon modules, and the plasmid modified by Gibson assembly to generate pCDH-EF1a-HygroR-T2A-TetOn3G with hygromycin resistance (for DExogron). Other plasmids used (for details see excel file S1): pLenti Lifeact-iRFP670 BlastR (*Padilla-Rodriguez et al., 2018*) (Addgene Plasmid #84385); pGIZ-PuroR-IRES-GFP_shRNA anti Rab11a/b (Dharmacon); GFP/mCherry-R11a/b (created by Davidson or Caswell lab); lentiviral plasmids coding two versions of PA-Tet-OFF/ON (*Yamada et al., 2018*). All constructs were verified first by restriction analysis and then by sequencing. Further details on the cloning strategy and plasmids used in this study are available from Addgene (number 179887–179904) or upon reasonable request.

## Cell culture

A2780-DNA3 (*Caswell et al., 2007*) and OVCAR-3 (ATCC) ovarian cancer cell lines were cultured in RPMI-1640 medium (R8758, Sigma). Ovarian cancer COV362 (ECACC), TIFs (telomerase immortalized fibroblasts *Caswell et al., 2007*), and HEK293Ts cells were cultured in Dulbecco's Modified Eagles Medium (DMEM) (D5796, Sigma). All cell culture medium was supplemented with 10% v/v fetal bovine serum (FBS) and ciprofloxacin (0.01 mg/ml; Sigma) and cells were maintained at 37°C in a humidified atmosphere with 5% (v/v) $CO_2$. Cells were confirmed as being negative for mycoplasma contamination by PCR. A2780 was selected for antibiotic resistance as follows: hygromycin 200 μg/ml; BlaR 5 μg/ml; PuroR 0.5–1 μg/ml.

## Lentivirus packaging

Lentiviral particles were produced in HEK293T cells, via a polyethylenimine-mediated transfection with packaging plasmids psPAX2 and pM2G with lentiviral vector pCDH or pLenti in the ratio 1.5:1:2. Similarly, retroviral pRetroQ was delivered with pM2G and specific gag-pol plasmid. Supernatants were collected at 94 hr after transfection, filtered through a 0.45 μm filter and added to target cells. Transduced cells were selected by appropriate antibiotics or grown up and sorted by FACS.

## Flow cytometry and analysis

Cells (pre-induced ±dox ± IAA as needed) were gated based on mNeonGreen/mCherry/mtagBFP expression as desired and bulk sorted into 15 ml centrifuge tubes by FACS Aria II (BD). In few cases, cells were cloned by serial dilution. Sorted DExCon/DExogron cells pre-induced by dox were again screened for fluorescence activation after 2 weeks of growing in the absence of dox. Flow cytometry was performed using an LSR Fortessa (BD) with a five-laser system (355, 405, 488, 561, and 640 nm) and fluorescence levels recorded with for 10,000 events with configuration as follows: mNeonGreen (488 530/30), mCherry (561 610/20), mtagBFP (405 450/50), and iRFP670 (640 670/14). For all experiments, cells were pre-treated ±dox ± IAA (24–72 hr) prior analysis and gated based on the empty A2780 (negative control). If analysis could not be done same day as cell preparation, cells were detached using PBS with 5 mM EDTA, fixed with 2% paraformaldehyde (PFA) on ice (10 min), and exchanged to 0.5% BSA in PBS supplemented with 0.1% sodium azide. Data were processed using the FlowJo software (FlowJo LLC). TFN recycling assay was carried out in six well plates (~90% confluence) as follows: 1× wash by PBS followed by serum starvation (30 min; 37°C) and pulsing 30 min (37°C) with 10 μg/ml of labeled TFN (Alexa Fluor 647 Conjugate; Thermofisher #T23366). Cells were then washed 2× on ice followed by chasing of TFN recycling in the excess of un-labeled Holo-TFN (100 μg/ml; sigma; #T0665) for a defined time (10; 30 or 60 min). Finally, cells were washed on ice 2× by ice cold PBS, suspended in 500 μl of PBS 5 mM EDTA, and transferred to tubes with 500 μl 4% PFA. After 10 min, cells were centrifuged (5 min, 500 g; 0°C) and transferred to 1 ml of Flow cytometry buffer (0.5% BSA, PBS, and 0.01% sodium azide) and analyzed by Flow cytometry as descibed above.

## Antibodies and reagents

The following antibodies and reagents were used (details in excel file S1): Rab11 (rabbit pAb, # PA5-31348), α-tubulin (mouse mAb DM1A, #ab7291), Rab11a (rabbit mAb D4P6P, #2,413S), Rab25 (rabbit pAb, #13,048S), RFP (rat mAb 5F8), and mNeonGreen (mouse mAb 32F6-100). Fluorescent secondary antibodies (Li-cor, Invitrogen or Jackson ImmunoResearch Laboratories) were used as recommended

by the manufacturer. The reagents used were BlaR S HCl (Gibco) and PuroR dihydrochloride (Thermo), hygromycin B (Roche), fibronectin (sigma), high density type I collagen (Corning), TFN (Alexa Fluor 647 Conjugate; Thermofisher #T23366), and Holo-TFN (sigma; #T0665); SiR-Tubulin (Cytoskeleton), Dynabeads MyOne Streptavidin C1 (Thermo), SuperScript IV Reverse Transcriptase (ThermoFisher), and Agencourt AMPure XP (Beckman Coulter). Doxycycline hydrochloride (Sigma) was dissolved as 250 µg/ml in water (stored –20°C in aliquiots) and used within 14 days if kept in 4°C. Similarly, IAA sodium salt (source of auxin; Cambridge Bioscience Limited, #16954–1 g-CAY) aliquiots (10 mg/ml in water) were in stored –20°C, but used within 2 days once melted.

## SDS-PAGE and quantitative western blotting

Cells were lyzed in denaturing lysis buffer (2% SDS, 20% glycerol, 120 mM tris pH 6.8, and 0.1% bromophenol blue) and heated 10 min 98°C. Cell lysates were resolved under denaturing conditions by SDS-PAGE (4–12% Bis-Tris gels; Invitrogen) and transferred to nitrocellulose membrane using Trans-Blot Turbo Transfer System (Bio-rad). Membranes were blocked with 4% BSA-TBS (Tris-buffered saline) for 1 hr followed by incubation overnight at 4°C with the appropriate primary antibody in 2–3% BSA TBST (TBS with 0.05% Tween 20) and then incubated for 1 hr with the appropriate fluorophore-conjugated secondary antibody in 2.5% milk-TBST. Membranes were scanned using an infrared imaging system (Odyssey; LI-COR Biosciences).

## Bioinformatic analysis

Rab11 expression levels (RNAseq, free of batch-effects) across different tissues and their relationship to cancer were analyzed using UCSC Xena https://xenabrowser.net/ (*Goldman et al., 2020*) from the TCGA TARGET genotype-tissue expression (GTEx) study and graphs generated in Prism software or visualized using Qlucore Omics Explorer. UCSC Xena tool was used to compare the methylation pattern of Rab25 and its expression (TCGA Pan-Cancer or GDC Pan-Cancer study). Computed quantitative trait loci, eQTLs, of Rab11s isoforms across multiple healthy tissues (https://doi.org/10.48420/16988617) were visualized using the GTEx browser: https://gtexportal.org/ (dbGaP accession number phs000424.vN.pN on 05/06/2021).

## Preparation of long ssDNA template

ssDNA was prepared through optimized reverse-transcription of an RNA intermediate (IVT) for sequences up to 2 kb using Hifi reverse transcriptase, similarly to that described in detail in *Li et al., 2017*. Briefly, all donor plasmids (details in excel file S1) with a T7 promoter site followed by sequences used for knock-in (fluorophore sequence flanked by 150–300 bp homologous arms) were first linearized by the appropriate restriction enzyme (cutting just outside the homologous arm opposite to the T7 promoter), followed by in vitro transcription (HiScribe T7 polymerase, NEB #E2040S), DNA degradation (TurboDNAse, Thermo #AM2238), reverse transcription (SuperScript IV, Thermo #18090050) with RNAse Inhibitor (SUPERase, AM2696 Thermo), and RNA hydrolysis (95°C 10 min in pH >10 with EDTA). Final ssDNA product and all intermediate steps were purified using SPRI beads (AMPure XP, Agencourt) according to manufacturer protocol. Longer ssDNA sequences (>2 kb) were then prepared using PCR with biotinylated forward primers (*Bennett et al., 2021*) (see diagram in excel file S1), final ssDNA purified using SPRI beads (AMPure XP, Agencourt), and anti-sense ssDNA strand used for knock-in. Where donor plasmid was used as a DNA template, biotinylated PCR-product was purified using Dynabeads MyOne Streptavidin C1 (Cat# 65001, Thermo) and anti-sense ssDNA strand eluted by 20 mM NaOH (later neutralized by HCl). All ssDNA were denatured by heating (70°C, 10 min) in RNA loading dye containing formamide (Thermo) prior to verification by agarose-gel electrophoresis and/or sequencing. Sequences of used primers are found in excel file S1 and detailed protocols can be download from https://doi.org/10.48420/16878859.

## RNA extraction and reverse transcription, qPCR

RNA was extracted using the Qiagen RNeasy Kit according to the manufacturer's instructions from A2780 cells growing in six-well plates. RNA (1 µg) was first incubated with Oligo(dT)12–18 (Invitrogen) primers and 20 µM dNTP at 70°C for 10 min, and placed back on ice followed by reverse transcription using 'M-MLV' reverse transcriptase (Promega, #M1705) together with 1 units/ml murine RNAse inhibitor (New England Biolabs ltd.) for 50 min at 37°C. All yields were diluted in RNAse-free water to

the same final concentration (200 ng/µl) prior to PCR or qPCR. Sample cDNA was added to PowerUp SYBR Green Master Mix (Applied Biosystems) with 500 nM primers (listed in excel file S1), and qPCR was performed in technical triplicate per sample with the following cycling conditions: 50°C for 2 min, 95°C for 2 min, followed by 35 cycles at 95°C for 15 s, 60°C for 15 s, and 72°C for 60 s. Amplification was carried out using Mx3005P qPCR System from Agilent Technologies. Relative gene expression was determined using ΔΔCt method relative to the average of ΔCt of 201 Rab11b isoform (same cDNA) and fold gene expression calculated using formula $2^{-(\Delta\Delta Ct)}$. PCR products were ran on a 2% agarose gel with SYBR green to check primer specificity and for quantification of band intensities.

## RNP assembly and magnetofection and validation

We modified, improved, and optimized a previously published simplified forward transfection protocol (*Jacobi et al., 2017*) by delivering Hifi Cas9 (Alt-R S.p. HiFi Cas9 Nuclease V3, IDT) and guideRNA (crRNA:tracrRNA) as RNP together (final 10 nM) with ssDNA donor (100–300 b homologous arms) via CrisperMAX (Thermo) combined with nanoparticles for magnetofection. Combimag nanoparticles (OZBiosciences) were used according to the manufacturer suggestions, with a magnetic plate. To increase HDR efficiency of shorter homologous arms used for Rab11b knock-in (100–150 bases), RAD51-stimulatory compound 1 (RS-1, *Song et al., 2016*) was successfully implemented in our protocol (20 µM prior magnetofection), but did not improve efficiency with longer arms used for Rab11a and Rab25 (300 bases). GuideRNA (crRNA:tracrRNA) cutting in a 10 bp proximity to the ATG insertion site was selected with a preference for low off target potential. Sequences of tracrRNA and donor plasmids used for ssDNA preparation, all designed to be resistant to Cas9 cleavage, are provided in excel file S1. The amount of long ssDNA (15–105 ng) was critical to optimize for each ssDNA preparation method as excess of highly pure full-length ssDNA led to profound cell death and was sometimes accompanied by non-specific knock-in events. Knock-in was verified by FP expression and localization, and FACS sorted cells validated for specific full-length knock-in by western blot and by sequencing of PCR-amplified genomic DNA. Primers binding outside the homologous arms used for knock-in were used together with mNeonGreen or mCherry specific primers followed by nested PCR with additional primers prior sequencing (sequences of primers are listed in excel file S1). Detailed protocol including screening can be download from https://doi.org/10.48420/16878859.

## Imaging, immunostaining, and colocalization

Low resolution widefield images were acquired using an Olympus IX51/TH4-200 optical microscope with UPLANFL N 4× /0.13 PHL or 20 ×/0.40 LCAch Infinity-Corrected PhC Phase Contrast Objective; mercury lamp (U-RFL-T power supply). The images were collected using a QImaging Retiga-SRV CCD digital camera and QCapture software. For fixed and live cell imaging, cells were grown in glass bottom culture dishes with #1.5 high performance cover glass coated with FN (10 µg/ml) or in µ-Plate 96 Well Black (#1.5 polymer, tissue culture treated, Ibidi Cat. #89626) and cells imaged in 1 × Opti-Klear medium (Marker Gene Technologies Inc) supplemented with 10% (v/v) FCS at 37°C. Fluorescence high-resolution timelapse images were acquired using a 3i Marianis system with CSU-X1 spinning disc confocal (Yokagowa) on a Zeiss Axio-Observer Z1 microscope with a 63 ×/1.40 Plan-Apochromat objective, Evolve EMCCD camera (Photometrics), and motorized XYZ stage (ASI). The 405, 488, 561, and 633 nm lasers were controlled using an AOTF through the laserstack (intelligent imaging innovations [3I]) allowing both rapid 'shuttering' of the laser and attenuation of the laser power. Images were captured using SlideBook 6.0 software (3i). When acquiring 3D optical stacks, the confocal software was used to determine the optimal number of Z sections. Maximum intensity projections of these 3D stacks are shown in the results. For colocalization analysis of Rab11a and Rab11b, the colocalization finder tool in ImageJ was used to generate Pearson's correlation coefficients. Plot profiles were generated by 'plot profile' function in ImageJ, normalized to maximal intensity (set to 1000) or by 0–1 scaling and visualized in Prism software. Kymographs were created from timelapse images using KymoToolBox in ImageJ. Four colors were visualized using BIOP Lookup Tables (spring green, amber, bright pink, and azure). For fixed cell imaging, cells were fixed in PFA (4%, 15 min), blocked by 1% BSA with 0.1–0.2% saponin in PBS followed by immunofluorescence staining in the same buffer using primary RFP (rat mAb 5F8; 1:200;) or mNeonGreen (mouse mAb 32F6-100; 1:200) and corresponding secondary antibodies (1:200 of anti-rat Alexa Fluor 594 or anti-mouse Alexa Fluor

488, Jackson ImmunoResearch Laboratories). Cells were mounted directly in μ-Plate 96 Well Black Ibidi plate using ProLong Gold Antifade Mountant (Thermo Fisher).

A Zeiss LSM 880 Airyscan confocal microscope was used for live imaging triple knock-in A2780 cells (mNeonGreen-Rab11a; mCherry-Rab11b; mTagBFP2-Rab25 DEXON±Alexa-647 labelled Transferrin =TFN-647) using a 63×/1.46 Plan-Apochromat objective and 1× confocal zoom with pinhole one airy unit. Individual emission spectra (mtagBFP2; mNeonGreen, mCherry, TFN-647) were first determined with a 34 channel spectral detector and 405/488/594/633 lasers, adjusted, and collected using an Airyscan detector to prevent any crosstalk (Scanning mode LineSequential; 1024 × 1024). These images were then loaded to Comdet plugin to simultaneously track all mNeonGreen/mCherry/mTabBFP2/TFN-647 positive vesicles (constant particle size: 6.0; Max distance between colocalized spots: 9.0; 100 cells from three independent experiments). These results were then re-calculated with respect to each channel using a custom-made Python script (provided in https://doi.org/10.6084/m9.figshare.16810546) to map all unique colocalization outcomes and manually visualized as a Venn diagram with the percentage of colocalization for every channel as 100% total. Images of triple knock-in cells recycling TFN-647, which were spread overnight on 5 mm diameter fibronectin-coated coverslips (live or fixed), were collected using a 3i Lattice Light Sheet microscope using a 25×/1.1 water dipping imaging objective with a combination of three diodes: 405/488/560/642 nm Bessel beam array (with 100% laser power and Bessel beam length of 50 μm and a ORCA Flash V4 CMOS camera [Hamamatsu] with 50 ms or 200 ms exposure time for mNeonGreen/mCherry/TFN-647 or mTagBFP2, respectively). The system was recalibrated for the refractive index of the 1× Opti-Klear medium (Marker Gene Technologies Inc) supplemented with 10% (v/v) FCS before imaging.

LUXon cells were blue light irradiated before imaging in $CO_2$ incubator with LED flashlight torch at 25 cm distance with high or low mode (high = 2420 lux [227 FC] and low=613 lux [56 FC]) for the specified time. siR Tubulin (Cytoskeleton, Cat# CY-SC002) was added 30 min before imaging (400 nM). For Alexa647-Transferrin labeling, cells were incubated 10 min on ice, washed with cold Opti-mem media, followed by 30 min incubation at 37°C with 25 μg/ml Alexa647-Transferrin, another Opti-mem wash, and a change to Opti-Klear Live Cell Imaging Buffer for immediate imaging. Hoechst (#3,342 Thermo) was also added (5 μg/ml) to Opti-Klear media, at least 1 hr before imaging (no washing).

## Knocksideways

DExCon modified mCherry-Nanobody-Rab11a/b/25 A2780 cells were pre-induced with dox for 72 hr and magnetofected with GFP-FKBP and pMito-iRFP670-FRB 24 hr prior to imaging directly in μ-Plate 96 Well Black (Ibidi Cat. #89626). Cells were imaged in 1× Opti-Klear medium (Marker Gene Technologies Inc) supplemented with dox (250 ng/ml) and 10% (v/v) FCS with the 3i Marianis CSU-X1 spinning disc system. Images were first taken for randomly chosen cells expressing constructs across multiple imaging areas before rap (200 nM, Sigma) addition. Images were captured at 20 s intervals for approximately 45 min, with the time between rap addition and timelapse initiation recorded. Timelapses were analyzed via ImageJ using custom-made macro automation (provided in https://doi.org/10.6084/m9.figshare.17085632) of the Colocalization Finder plugin to evaluate the Pearson's correlation coefficient between mCherry and mitochondrial (pMito) signal over defined areas containing transformed cells using a consistent proportional threshold; this data was expressed both as a series of colocalization graphs plotting the relative intensities of each channel for each pixel as well as a numerical Pearson's correlation coefficient value. The individual datasets per timelapse were processed by custom outlier detection (provided in https://doi.org/10.6084/m9.figshare.17085632) utilizing a Gaussian analysis of differences between data points to split the data into subsets of congruent data points, which were then reconnected via subset congruency to produce the single prevailing set of data without outliers. Individual timelapses were scaled per their range to normalize differences between datasets for comparison. This data was then processed as whole data aggregates and individual cells used to generate one-phase association curve fits, minimum and maximum set at 0 and 1 respectively, using Prism software. Calculated halftimes of Rab11 family members generated through individual curve fits were tested for significant differences via one way ANOVA with Tukey's multiple comparisons.

## High throughput imaging and expression/degradation kinetics

Incucyte S3 Live Cell Analysis system, situated in temperature and $CO_2$ controlled incubator, was used for high throughput imaging. Images of cells in phenol red-free RPMI media (Thermo, #11835030)

supplemented with 10% v/v FBS and ciprofloxacin (0.01 mg/ml; Sigma) were acquired with a 10×/1.22 Plan Fluor (wound healing, proliferation) or 20×/ 0.61 S Plan Fluor (fluorescence kinetics) objective and the green and red filter set for fluorescence images with standard exposure settings. Wound healing: confluent A2780 cells seeded on 96-well ImageLock microplates were pre-treated for 24±dox (250 ng/ml)±IAA (100 µg/ml), scratched by the WoundMaker to create homogeneous 700–800 µm wide wounds and automatically imaged every 30 min or 1 hr in RPMI phenol-free media ±dox/IAA using Incucyte S3 system. This system automatically detected and quantified the ratio of the cell-occupied area to the total area of the initial scratched region (phase-contrast). This data was then processed as whole data aggregates and individual wells (n=6–9 across three replicates) were fitted by one-phase association function (Prism software) to calculate the time needed to close 50% of the normalized wound area. Cell proliferation was analyzed on Costar 96 flat bottom plates (Corning 3596) by Incucyte software trained to recognize cell occupied area (=confluence %) based on the brightfield images taken every 30 min and its change over 72 hr. At time 0, cells were seeded sparsely (5–10 thousand cells/well)±dox (250 ng/ml)±IAA (100 µg/ml) and the logarithm of two calculated from the cell confluence normalized to 5 hr (starting point of potential dox/IAA effect). Curves slopes were then obtained by linear regression from the linear part of the individual curves (5–40 hr; n=6–9 across three replicates) and used for calculation of the doubling time [T] by the formula T=log(2)/slope. For fluorescence expression/degradation kinetics analyzed by Incucyte, approximately 20,000 cells were seeded per well of Costar 96 flat bottom plate (Corning 3596)±dox/IAA and immediately imaged (red/green/bright) every 30 min (first 2 hr every 15 min) up to 72 hr. Fluorescence intensity was automatically analyzed as integrated fluorescence intensity per image divided by phase area confluence change to count for the effect of cell proliferation. To calculate halftime of degradation (expression) kinetics induced by IAA treatment (Dox), data were fitted by one-phase association function. Where appropriate, individual values were scaled per their range as to normalize differences between datasets for comparison (minimum and maximum set at 0 and 1000 respectively).

A BioTek Synergy H1 microplate reader was used to analyze expression kinetics or protein stability of dox-preinduced cells seeded confluent (100% monolayer) in Costar 96 flat bottom plate (Corning 3596)±dox with Leibovitz's L-15 phenol red-free ($CO_2$ independent media; #21083027 Thermofisher) supplemented with 10% v/v FBS and ciprofloxacin (0.01 mg/ml). Readings were captured every 1 hr over 24–48 hr at 37°C (gain 100 399/454; gain 100 490/517 nm; gain 150 580/610 nm). A2780 cells were growing in L-15 media only up to a monolayer and differences in cell numbers were during readings negligible (mTagBFP fluorescence of Teton3G-T2A- mTagBFP). To be able to monitor fluorescence change over 72 hr, cells were seeded 24 hr before the experiment ±dox and fluorescence intensity measured over the following 48 hr. To account for the phototoxic effect of fluorescence readings and compare differences between datasets, fluorescence kinetics were scaled per their range (minimum and maximum set at 0 and 1000 respectively). Data were aggregated and fitted by one-phase association function (to calculate halftime) or by polynomial function.

Statistics were calculated from halftimes generated through individual curve fits (expression kinetics) or slopes derived from the linear part of the curve (protein stability), unless otherwise indicated.

## CDM and 3D invasion assays

CDMs were generated as previously described (*Caswell et al., 2007*; *Cukierman et al., 2001*). Briefly, plates were coated with 0.2% gelatin (v/v, Sigma Aldrich), crosslinked with 1% glutaraldehyde (v/v, Sigma Aldrich), and quenched with 1 M glycine (Thermo Fisher) before TIFs were confluently seeded. DMEM medium supplemented with 25 µg/ml ascorbic acid (v/v, Sigma Aldrich) was changed every 48 hr for 8 days. Cells were denuded with extraction buffer (0.5% v/v) Triton X-100; 20 mM ammonium hydroxide ($NH_4OH$) to leave only matrix. Finally, phosphodiester linkages in the DNA backbone were cleaved by DNAse I (Lonza).

Collagen/FN matrix for invasion assays was prepared as follows: Collagen solution (10 mg/ml, Corning #354249) was diluted to the final mix: 2.25 mg/ml, 1×RPMI, 15 mM HEPES (750 mM), 8.5 mM NaOH (1 M), 0.4% NaHCO3 (7.5 %, Sigma), 5 µg/ml folic acid, and supplemented with 25 µg /ml fibronectin (labeled or un-labeled as needed). FN was dialyzed into PBS by Slide-A-Lyzer Dialysis Cassette (Thermo) and conjugated with Alexa-fluor 647 NHS ester (Thermo). For LUXon invasion experiments Alexa-647 conjugated FN was then supplemented in a 1:10 ratio (labeled/non-labeled, total 25 µg/ml). For the modified 3D cell-zone exclusion assay (*Gemperle et al., 2019*), 50 µl of the

collagen/FN (un-labeled) mix was added into well of a 96-well plate and polymerized at 37°C. Cell suspension (100 µl; $1\times10^6$ cells/ml) was added on top of the collagen/FN gel. After cell attachment (4 hr), the medium was removed and scratch was performed. Immediately after removal of the rest of media in the generated wound, another layer of collagen/FN mix (100 µl, including FN-647) was added (schematic of the experiment is shown in 3I). Collagen was allowed to polymerize for 30 min at RT/37°C, RPMI medium added, and cells incubated for 24 hr at 37°C in a humidified 5% (v/v) CO2 atmosphere to allow invasion. For spatiotemporal re-activation of Rab25 expression, wells were partly illuminated by blue LED for first 9 hr (black plasticine was used to block light on 80% of the well area).

Spheroids of uniform size were prepared using MicroTissues 3D Petri Dish micro-mold spheroids (microtissues; #12–81) for on-chip spheroid invasion assays. Briefly, 600 µl of 2% UltraPure Agarose (v/v in PBS; Sigma) was allowed to form gels in 3D petri dish micro-mold with 81 circular recesses (9 × 9 array; 800 µm each in diameter). 40,500 cells/190 µl (500 cells/spheroid with an estimated diameter 100 µm) were introduced to the micro-mold and allowed to self-assemble into spheroid over 48 hr. To increase the compactness of the spheroid, 10% serum was changed to 1% on the second day of spheroid formation (±dox/IAA). Medium was fully aspirated and 190 µl of collage/FN introduced and polymerized at 37°C. Chip with 81 spheroids embedded in collagen/FN gel were overlayed with 0.8–1% UltraPure Agarose (v/v in PBS) to prevent gel movement and shrinking. Finally, chips were overlayed with RPMI media (10% serum;±dox/IAA 2 × excess to count for the volume of agarose), imaged (4 × objective, phase contrast) and cells allowed to invade for 48–72 hr at 37°C in a humidified 5% (v/v) $CO_2$ atmosphere (±blue light illumination) before additional imaging. Detailed protocol can be downloaded from https://doi.org/10.48420/16878859. Cell invasion area was calculated custom-written macro code for ImageJ that automatically created a binary mask of invading cells and subtracted the initial spheroid area (provided in https://doi.org/10.48420/16878829).

## Statistical analysis

Data were tested for normality and one-way ANOVA with Tukey post hoc test used for multiple comparisons or as indicated in legend. Where data were not normally distributed, ANOVA on ranks was used instead. All statistical analysis has been performed with GraphPad Prism software, where *** denotes $p<0.001$, ** denotes $p<0.01$, and * denotes $p<0.05$. Data represent at least three independent experiments; n numbers and p values are described in relevant figure legends.

## Data availability

List of antibodies, chemicals, materials, primers, and all necessary sequences for replication of this study are provided (excel file S1 or accessible from https://doi.org/10.48420/14999550), along Graph Prism files (raw or normalized data at https://doi.org/10.48420/16878904), detailed innovative protocols (https://doi.org/10.48420/16878859), and custom scripts/macro codes are deposited on Figshare (links in the methods section). Annotated plasmid maps used and generated in this study (including catalog of donors) can be downloaded from https://doi.org/10.48420/16810525 and 10.48420/14999526. Plasmids can be provided upon reasonable request from corresponding author or will be available from Addgene. Vidoes can be downloaded from https://doi.org/10.48420/17111864.

## Acknowledgements

We thank Hayley Bennett for providing a protocol for long ssDNA preparation using biotinylated primers and all of the members of the Caswell lab for their support, especially Eleanor Hinde. We also are grateful to Itaru Imayoshi and Mayumi Yamada for providing lentiviral plasmids coding two versions of PA-Tet-OFF/ON. The Bioimaging Facility microscopes used in this study were purchased with grants from BBSRC, Wellcome, and the University of Manchester Strategic Fund. We thank Peter March, Steven Marsden, and Dave Spiller for their help with microscopy. We further thank the University of Manchester Flow Cytometry Core Facility for assistance with flow cytometry and sorting. The Flow Cytometry Core Facility is supported, in part, by the University of Manchester with assistance from MRC Grant ref MR/L011840/1. This project received funding from the European Union's Horizon 2020 research and innovation program under grant agreement No (836,212), the MRC (MR/R009376/1), and CRUK (DCRPGF\100002; CF is funded by the Cancer Research UK Manchester Centre [C147/A25254] Non-Clinical Training Program) and the Wellcome Trust Centre for Cell Matrix

research is funded by grant 203128 /A/16/Z. GTEx project was supported by the Common Fund of the Office of the Director of the National Institutes of Health, and by NCI, NHGRI, NHLBI, NIDA, NIMH, and NINDS.

## Additional information

### Funding

| Funder | Grant reference number | Author |
|---|---|---|
| Cancer Research UK | DCRPGF\100002 | Jakub Gemperle<br>Thomas S Harrison<br>Patrick T Caswell |
| Horizon 2020 Framework Programme | 836212 | Jakub Gemperle<br>Patrick T Caswell |
| Cancer Research UK | C147/A25254 | Chloe Flett |
| Medical Research Council | MR/R009376/1 | Patrick T Caswell |
| Wellcome Trust | 203128/A/16/Z | Jakub Gemperle<br>Thomas S Harrison<br>Chloe Flett<br>Antony D Adamson<br>Patrick T Caswell |

The funders had no role in study design, data collection and interpretation, or the decision to submit the work for publication. For the purpose of Open Access, the authors have applied a CC BY public copyright license to any Author Accepted Manuscript version arising from this submission.

### Author contributions
Jakub Gemperle, Conceptualization, Data curation, Formal analysis, Funding acquisition, Investigation, Methodology, Resources, Software, Supervision, Writing - original draft, Writing - review and editing; Thomas S Harrison, Data curation, Formal analysis, Investigation, Methodology, Software, Writing - review and editing; Chloe Flett, Formal analysis, Investigation, Methodology, Writing - review and editing; Antony D Adamson, Conceptualization, Methodology, Writing - review and editing; Patrick T Caswell, Conceptualization, Data curation, Formal analysis, Funding acquisition, Methodology, Project administration, Resources, Supervision, Writing - original draft, Writing - review and editing

### Author ORCIDs
Jakub Gemperle http://orcid.org/0000-0001-8360-7075
Thomas S Harrison http://orcid.org/0000-0003-2938-1831
Chloe Flett http://orcid.org/0000-0002-0950-3128
Patrick T Caswell http://orcid.org/0000-0002-2633-2324

### Decision letter and Author response
Decision letter https://doi.org/10.7554/eLife.76651.sa1
Author response https://doi.org/10.7554/eLife.76651.sa2

## Additional files

### Supplementary files
• Transparent reporting form

• Source data 1. Annotated uncropped blots. Annotated uncropped blots for *Figures 1–3*, *Figure 5*, *Figure 6*, *Figure 1—figure supplement 2*, *Figure 2—figure supplement 1*, *Figure 3—figure supplement 1*, *Figure 4—figure supplement 1*, *Figure 6—figure supplements 1 and 2*, *Figure 6—figure supplement 4*.

### Data availability
Code and analysis tools developed are freely available via Figshare.

The following datasets were generated:

| Author(s) | Year | Dataset title | Dataset URL | Database and Identifier |
|---|---|---|---|---|
| Harrison T | 2022 | Tools for temporal knocksideways analysis | https://doi.org/10.6084/m9.figshare.17085632 | figshare, 10.6084/m9.figshare.17085632 |
| Gemperle J | 2022 | GraphPrism files with raw/normalized data supporting DExCon study | https://doi.org/10.48420/16878904 | Figshare, 10.48420/16878904 |
| Gemperle J | 2022 | Macro for spheroid invasion assay including protocol | https://doi.org/10.48420/16878829 | figshare, 10.48420/16878829 |
| Harrison T | 2022 | Colocalisation Quantifier Python Script | https://doi.org/10.6084/m9.figshare.16810546 | figshare, 10.6084/m9.figshare.16810546 |

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
