## [Editor Report]

Here the authors present a genome editing strategy that enables blocking and tetracycline-controlled, re-expression of fluorescently-tagged genes from endogenous loci. The authors combine this with a photoactivatable, tet-on/off system, a knocksideways approach, and the auxin-inducible degron system to improve spatial and temporal control of gene expression. These powerful tools are used to evaluate the localization, function and protein-expression dynamics of the Rab11 family of small GTPases.

---

## [Decision Letter]

**Decision letter after peer review:**

Thank you for submitting your article "DExCon, DExogron, LUXon: on-demand expression control of endogenous genes reveals differential dynamics of Rab11 family members" for consideration by *eLife* in the category of TOOLS and RESOURCES. Your article has been reviewed by 2 peer reviewers, and the evaluation has been overseen by a Reviewing Editor and Suzanne Pfeffer as the Senior Editor. The following individual involved in review of your submission has agreed to reveal their identity: Rytis Prekeris (Reviewer #2).

Essential revisions:

1. Either compare various Rab11 members with regard to recycling of different integrin receptors (that have different recycling kinetics) in the available cells [Or] identify the Rab11 splice variants expressed in A2780 cells and demonstrate that the relative abundance of these variants is not altered upon fluorescent-tagging and CMV-promotor-driven overexpression of Rab11 from the endogenous locus.

2. Please show that the Rab11a/b/Rab25 expression levels used do not alter localization and function. This could be tested simply by a transferrin recycling assay. To ensure that DExCon-Rab11/25 expression levels do not affect localization, the authors could use cells containing a knock-in of mCH-Rab11a on one allele and DExCon-mNG-Rab11a on the other allele and compare their localization.

3. Please repeat the experiment 6F under -dox +IAA conditions.

*Reviewer #2 (Recommendations for the authors):*

It would be very interesting and perhaps more informative to analyze Rab11 family in more interesting cellular contexts (instead of transferrin recycling in cancer cells). While re-tagging Rab11a, Rab11b and Rab25 in other cell lines is not really practical, using a few other cargoes (that are differentially regulated) would substantially enhance the paper. For example, comparing various Rab11 members and recycling of a few different integrin receptors (that have different recycling kinetics) would be very interesting, especially since transferrin is bulk recycled and does not have very intricate regulation that is would be differentially affected by Rab11a, Rab11b or Rab25.

---

## [Author Response]

Essential revisions:1. Either compare various Rab11 members with regard to recycling of different integrin receptors (that have different recycling kinetics) in the available cells [Or] identify the Rab11 splice variants expressed in A2780 cells and demonstrate that the relative abundance of these variants is not altered upon fluorescent-tagging and CMV-promotor-driven overexpression of Rab11 from the endogenous locus.

We identified two Rab11b splice variants that could produce protein coding products of similar molecular weight to the single band we see in A2780 cells by western blotting (201 (24.5kDa) and 202 (20kDa)). Using qPCR and semi quantitative rtPCR we determined that the relative abundance of 201 and 202 splice variants is not significantly altered by the DexCon module. This was confirmed with a third primer combination that detects, another predicted splice variant (204, likely untranslated due to an early stop codon) in addition to 201 and 202 and demonstrated that the relative ratio of splice variants is not altered upon fluorescent-tagging and CMV-promotor-driven expression of Rab11 from the endogenous locus, although there was slight increase in expression of all three variants from the dox-induced DexCon module. These data are included in a new Figure 2 Figure supplement 1B and described on page 5. For Rab11a, the splicing pattern is more complex, and we were unable to design specific primers to discern the major isoform (201, corresponding to full length ~24.4kDa Rab11a) from other splice variants (all of which generate shorter proteins that lack sequences essential to GTPase function). Unfortunately, we did not have time to test recycling of different integrins within the timeframe.

2. Please show that the Rab11a/b/Rab25 expression levels used do not alter localization and function. This could be tested simply by a transferrin recycling assay. To ensure that DExCon-Rab11/25 expression levels do not affect localization, the authors could use cells containing a knock-in of mCH-Rab11a on one allele and DExCon-mNG-Rab11a on the other allele and compare their localization.

These have been really useful suggestions, and we have included new data to demonstrate that Rab11a/b/Rab25 expression levels under DExCon control do not alter localization and that tuning of expression levels is important to maintain transferrin recycling. For localisation, we introduced the DExCon-mNeonGreen module into the second allele in cells where the first allele had a conventional mCherry-Rab11a or mCherry-Rab11b allele, introduced DExCon-mNeonGreen into one Rab11b allele and DExCon-mCherry into the second allele in the same cells, and introduced DExCon-mNeonGreen into Rab11a and DExCon-mCherry into Rab11b in the same cells before analysing their localisation using spinning disk confocal imaging (new Figure 5 figure supplement 2). In all CRISPR/Cas9 modified cells the localisation was near identical, although slight differences in the distribution of DExCon-mNeonGreen-Rab11a and DExCon-mCherry-Rab11b were again observed. These data are described in the text on page 13 as below:

“We further explored the possibility of simultaneous visualization of different alleles of the same gene (Figure 5E; Figure 5—figure supplement 1F, Figure 5—figure supplement 2A-C). We were able to successfully modify a second allele of mCherry-Rab11a, mCherry-Rab11b or DExCon-mCherry-Rab11b modified cells by knocking in DExCon-mNeonGreen modules (Figure 5F; Figure 5—figure supplement 1F,2A-C). Dox-mediated induction led to consistent almost identical localization in all cells with alleles modified with mCherry and DExCon-mNeonGreen for both Rab11a and Rab11b (Figure 5—figure supplement 2A-C). Consistent with previous results (Figure 2E; Figure 2—figure supplement 2E), DExCon-mNeonGreen-Rab11a was slightly more diffuse than DExCon-mCherry-Rab11b in the same cells (Figure 5—figure supplement 2D). This demonstrates that DExCon-induced Rab11 expression levels do not appreciably alter localization of the GTPases and generate signals compatible with routine imaging.”

For function of the DExCon-expressed Rab11a/b/25, we see clear positive effects on cell migration in 2D and in 3D matrix, which suggests these proteins are fully functional (when expressed at an appropriate level; Rab25: Figure 3K-N); (Rab11a/b 2D migration Figure 6F, 3D migration Figure 6G-I). We also tested the influence of Rab11a, Rab11b and Rab25 expression on TFN-R recycling. We found that the DExCon/DExogron induced knockout of Rab11a/b did not appreciably influence TFN-R recycling. This is consistent with the majority of TFN-R being recycled by the fast Rab4-dependent pathway, as is the case in other cell lines (e.g. MDCK cells, DOI: 10.1083/jcb.145.1.123). Indeed, inducing high level expression of Rab11a/b lowered the level of TFN-R recycling (new Figure 7B), consistent with the literature (DOI: 10.1073/pnas.95.11.6187). This effect was also notable at an early recycling timepoint (new Figure 7C), suggesting that this high expression level can divert TFN-R from fast to slow recycling routes. This is addressed in the text on page 20/21 as below:

“Recycling of TFN-R is typically biphasic with an initial rapid direct route from early endosomes to the plasma membrane (under control of Rab4) and a slower non-essential route that requires additional trafficking through perinuclear recycling endosomes via Rab11^32–34^ (Figure 7A). Our previous experiments demonstrated that the DExCon/Dexogron system induces expression of Rab11 family members to promote cell motility (Figure 3N; Figure 6F-H), the expressed proteins show endogenous localization within cells (2E; Figure 5—figure supplement 2A-D) and they traffic together with TFN (Figure 2—figure supplement 2D; Video 3), suggesting that they are fully functional. We further tested how DExCon/DExogron controlled Rab11 family expression level influenced recycling of internalised TFN to test the function of these re-expressed proteins in a different context. DExCon or Dexogron modified Rab11a or Rab11b cells which lack expression of the targeted GTPase(s) had little impact on TFN-R recycling at 30 mins (Figure 7B), suggesting that fast recycling is likely predominant in A2780 cells as in other cell types^34^. Re-expression of Rab25 did not impact upon recycling, indicating that this Rab11 family member may have a limited influence on trafficking this cargo (Figure 7B). However, dox induced high level expression of either Rab11a or Rab11b under DExCon/Dexogron control decreased the level of recycling at 30 minutes (Figure 7B), consistent with the literature^35^, perhaps by favouring the slow recycling route. DExCon/DExogron Rab11a/b cells selected for expression levels comparable to endogenous Rab11a/b knock-in did not have any significant effect on TFN-R recycling rate compared to the non-modified control, indicating that it is the expression level that slows recycling. Similarly, IAA/Dox co-treatment of DExogron-Rab11b cells led to near-physiological levels of Rab11b which had no significant effect on TFN-R recycling (Figure 7B). Surprisingly, TFN-R recycling slowed by high Rab11a DExCon levels was restored to the level of double knockout cells by lowering Rab11b levels with DExogron (Rab11a DExCon/Rab11b DExogron, IAA/dox co-treatment). These data suggest that Rab11a/Rab11b can recycle TFN-R via the slow recycling pathway, and that increasing levels of Rab11a/b could favour the slow recycling route to delay recycling of TFN.

To test the effect of an acute decrease of Rab11a/Rab11b levels towards the expression driven by endogenous locus, we pre-adapted A2780 cells to high Rab11a/Rab11b levels by treating them with Dox over >7 days before decreasing Rab11 levels by switching to IAA; dox+IAA or dox positive media for 24-48h prior analysis. TFN-R recycling was assessed after 10 minutes to highlight competition between fast and slow recycling routes (Figure 7C). Cells induced to have high level expression of DExCon-mNeonGreen-Rab11a or/and DExogron-mCherry-Rab11b again showed lower levels of TFN-R recycling compared to control (mainly evident for Rab11a), but tempering expression of Rab11a and Rab11b (or Rab11b alone in DExogron-mCherry-Rab11b cells) by withdrawing dox/adding IAA demonstrated that lower Rab11 levels correlated with increased recycling rates (7C). These data further support the notion that competition between fast and slow recycling likely controlled by Rab4 and Rab11a/b respectively underpins the differences in recycling rates induced by increasing Rab11 expression level.

Taken together, this suggests that the DExCon/DExogron system affords control over the timing and level of gene expression of Rab11 family members, and points towards a more complex relationship between fast Rab11-independent and slow Rab11-dependent recycling.”

3. Please repeat the experiment 6F under -dox +IAA conditions.

This is an important point and we have addressed this in a new Figure 6 figure supplement 2F and in the text on page 17 as follows:

“Because IAA induced depletion of DExogron Rab11b was incomplete when expression was driven in the presence dox, we pre-induced DExogron-Rab11b for 24 hours (Figure 6—figure supplement 2F) and then transferred cells to dox-free media with IAA (10 g/ml) for 24 hours before scratch. As expected, this condition led to complete mCherry removal (not detectable by fluorescent microscopy) and was sufficient to reverse the ‘rescued’ motility of DExogron-mCherry-Rab11b or DExogron-mCherry-Rab11b/DExCon-mNeonGreen-Rab11a cells (Figure 6—figure supplement 2F).”

Reviewer #2 (Recommendations for the authors):It would be very interesting and perhaps more informative to analyze Rab11 family in more interesting cellular contexts (instead of transferrin recycling in cancer cells). While re-tagging Rab11a, Rab11b and Rab25 in other cell lines is not really practical, using a few other cargoes (that are differentially regulated) would substantially enhance the paper. For example, comparing various Rab11 members and recycling of a few different integrin receptors (that have different recycling kinetics) would be very interesting, especially since transferrin is bulk recycled and does not have very intricate regulation that is would be differentially affected by Rab11a, Rab11b or Rab25.

We agree with this comment, but for the purposes of the revision we chose to focus on the technical aspects of the approach instead. We are certainly interested in pursuing this in more cancer appropriate cell lines in future.